# Offline Reinforcement Learning of
# High-Quality Behaviors Under Robust Style Alignment

**Mathieu Petitbois** [1 2]   **Rémy Portelas** [1]   **Sylvain Lamprier** [2]

## Abstract

We study offline reinforcement learning of style-conditioned policies using explicit style supervision via subtrajectory labeling functions. In this setting, aligning style with high task performance is particularly challenging due to distribution shift and inherent conflicts between style and reward. Existing methods, despite introducing numerous definitions of style, often fail to reconcile these objectives effectively. To address these challenges, we propose a unified definition of behavior style and instantiate it into a practical framework. Building on this, we introduce Style-Conditioned Implicit Q-Learning (SCIQL), which leverages offline goal-conditioned RL techniques, such as hindsight relabeling and value learning, and combine it with a new Gated Advantage Weighted Regression mechanism to efficiently optimize task performance while preserving style alignment. Experiments demonstrate that SCIQL achieves superior performance on both objectives compared to prior offline methods. Code, datasets and visuals are available in: `https://mathieu-petitbois.github.io/projects/sciql/`.

## 1. Introduction

A task can often be performed through diverse means and approaches. As such, while the majority of the sequential decision making literature has focused on learning agents that seek to optimize task performance, there has been a growing interest in the development of diverse agents that display a variety of behavioral styles, which can be crucial in human-robot interaction, autonomous driving and video games. While many previous works tackled diverse policy learning by relying on online interactions (Nilsson & Cully, 2021; Wu et al., 2023), the widespread availability of pre-recorded diverse behavior data (Hofmann, 2019; Mahmood et al., 2019; Zhang et al., 2020; Fu et al., 2021; Lee et al., 2024a; Jia et al., 2024; Park et al., 2025) catalyzed much progress in the learning of policies from such data without further environment interactions, allowing the training of high-performing agents in a more sample-efficient, less time-consuming and safer way (Levine et al., 2020). Such methods can be divided into two categories: Imitation Learning (IL) methods (Pomerleau, 1991; Florence et al., 2021; Chi et al., 2023) mimic expert trajectories, while offline Reinforcement Learning (RL) methods (Kumar et al., 2020; Kostrikov et al., 2022; Fujimoto & Gu, 2021; Chen et al., 2021; Nair et al., 2021; Garg et al., 2023) target high-performing behaviors based on observed rewards. Although some recent work has focused on diverse policy learning in both offline IL (Zhan et al., 2020; Yang et al., 2025) and offline RL (Mao et al., 2024), several challenges remain.

**Challenge 1: Style definition.** Literature dealing with style alignment ranges from discrete trajectory labels (Zhan et al., 2020; Yang et al., 2025) to unsupervised clusters (Mao et al., 2024) and continuous latent encodings (Petitbois et al., 2025), with distinct trade-offs: unsupervised definitions are often uncontrollable and hard to interpret, while supervised ones rely on manual labels and incur significant labeling costs. Additionally, since play styles span multiple timescales, attributing each local step to a style is non-trivial and can contribute to credit assignment problems. Furthermore, depending on the definition of style, assessing the alignment of an agent's behavior with respect to a target style may be difficult, which complicates alignment measurement and hinders policy controllability. As such, a key challenge is to derive a **general** definition that addresses **interpretability**, **labeling cost**, **alignment measurement**, and **credit assignment**.

**Challenge 2: Addressing state-style distribution shift.** While offline IL and offline RL are known to suffer from distribution shift due to environment stochasticity and compounding errors (Levine et al., 2020), the addition of style conditioning can exacerbate the issue by creating mismatches at inference time between visited states and target

[1]Ubisoft La Forge, Bordeaux, France [2]Université d'Angers, Angers, France. Correspondence to: Mathieu Petitbois <mathieu-petitboispt@gmail.com>.

*Proceedings of the 43rd International Conference on Machine Learning*, Seoul, South Korea. PMLR 306, 2026. Copyright 2026 by the author(s).

styles. For instance, a running policy may trip and fall into an out-of-distribution state-style configuration without the ability to recalibrate. While some previous work addressed this issue (Petitbois et al., 2025), most of them lack mechanisms to perform robust style alignment. Consequently, an open question is how to achieve **robust style alignment** without relying on further environment interactions.

**Challenge 3: Solving task and style misalignment.** Style alignment and task performance are often incompatible. For instance, a crawling policy may not achieve the same speed as a running one. Optimizing conflicting objectives of style alignment and task performance has been explored in offline RL, either by directly seeking compromises between them (Lin et al., 2024b;c; Yuan et al., 2025), or by shifting optimal policies from one objective to the other (Mao et al., 2024), but always at the cost of style alignment. Consequently, ensuring **robust style alignment while optimizing task performance** remains an open problem.

In this paper, we address these challenges through the following contributions:

**(1)** We propose a novel **general** view of the stylized policy learning problem as a generalization of the goal-conditioned RL (GCRL) problem (Park et al., 2025).

**(2)** We instantiate our definition within the supervised data-programming framework (Ratner et al., 2016) by using labeling functions as in Zhan et al. (2020); Yang et al. (2025) but on trajectory windows rather than full trajectories, capturing the multi-timescale nature of styles and mitigating high **credit assignment** challenges by design. The use of labeling functions also allows users to **quickly** program various **meaningful** style annotations for both training and evaluation, simplifying the **alignment measurement**.

**(3)** We introduce Style-Conditioned Implicit-Q-Learning (SCIQL), a style-conditioned offline RL algorithm inspired by IQL (Kostrikov et al., 2022) which leverages advantage signals to guide the policy toward the activation of target styles, making efficient use of **style-relabeling** (Petitbois et al., 2025) and trajectory stitching (Char et al., 2022) to allow for **robust style alignment**.

**(4)** Using the casting of the stylized policy learning problem as an RL problem, we introduce the notion of **Gated Advantage Weighted Regression (GAWR)** by using advantage functions as gates to allow **style-conditioned task-performance optimization**.

**(5)** We demonstrate through a set of experiments on diverse stylized RL tasks that our method effectively outperforms previous work on both **style alignment** and **style-conditioned task-performance optimization**. Code, datasets and visuals are available in: https://mathieu-petitbois.github.io/projects/sciql/.

## 2. Related Work

**IL and offline RL.** Imitation Learning methods seek to learn policies by mimicking expert demonstrations, usually stored as trajectory datasets, and can be grouped into different categories, including Behavior Cloning, classical Inverse RL (IRL), and Apprenticeship / Adversarial IRL. Behavior Cloning (BC) (Pomerleau, 1991) performs supervised regression of actions given states from the trajectory datasets, but suffers from compounding errors and distribution shifts (Ross et al., 2011). Classical IRL (Ng & Russell, 2000; Fu et al., 2018; Arora & Doshi, 2020) infers a reward under which the demonstration policy is optimal to optimize it via online RL. It is robust to distribution shifts but requires environment interactions. Apprenticeship / Adversarial IRL (e.g., GAIL (Ho & Ermon, 2016)) learns policies directly via implicit rewards, combining IRL's robustness with BC's direct learning, but typically requires online interactions. On the other hand, offline RL requires neither optimal demonstrations nor access to the environment. It uses reward signals to train policies offline and tackles distribution shifts via sequence modeling (Chen et al., 2021), biased BC (Nair et al., 2021; Fujimoto & Gu, 2021), policy conservativeness (Kumar et al., 2020), expectile regression (Kostrikov et al., 2022), or Q-value exponential weighting (Garg et al., 2023). In this work, we leverage offline RL techniques to jointly optimize behavior styles and task performance from reward signals, without assuming demonstration optimality.

**Diverse policy learning.** Capturing diverse behavior from a pre-recorded dataset has been addressed in the literature under various scopes. Several methods aim to capture a demonstration dataset's multimodality at the action level through imitation learning techniques (Florence et al., 2021; Shafiullah et al., 2022; Pearce et al., 2023; Chi et al., 2023; Lee et al., 2024b) while other methods aim to learn higher-timescale behavior diversity by learning to capture various behavior styles in both an unsupervised and supervised approach. In the IRL setting, InfoGAIL (Li et al., 2017), Intention-GAN (Hausman et al., 2017) and DiverseGAIL (Wang et al., 2017) aim to identify various behavior styles from demonstration data and train policies to reconstruct them using IRL techniques. Tirinzoni et al. (2025) aim to learn a forward-backward representation of a state successor measure (Dayan, 1993; Touati & Ollivier, 2021) to learn through IRL a policy that optimizes a wide variety of rewards with a bias toward a demonstration dataset. Also in the online RL setting, Wen et al. (2025) model the stylized policy learning problem as a constrained MDP problem to imitate style under the constraint of near-optimal task performance, while we tackle offline learning, prioritize style alignment over task performance and introduce relabeling strategies for robustness. In a BC setting, WZBC (Petitbois et al., 2025) learns a latent space of trajectories to employ trajectory-similarity-weighted-regression to im-

prove robustness to compounding errors in trajectory reconstruction. Further, SORL (Mao et al., 2024) learns a set of diverse representative policies through the EM algorithm and enhances them to perform stylized offline RL. Additionally, Lin et al. (2024a) introduce a playstyle similarity measure by extending the playstyle distance based on VQ-VAE state categories proposed in Lin et al. (2021), incorporating multiscale discretization, a perceptual kernel, and intersection-over-union weighting. We view our work and Lin et al. (2024a) as complementary. On the one hand, the VQ-VAE categories in Lin et al. (2024a) can serve as learned perceptual decision-point abstractions of states, providing a practical way to generate semantically meaningful labels in complex observation spaces such as pixel inputs. On the other hand, SCIQL can address the disjunction between dataset categories highlighted in Lin et al. (2024a), as it can stitch and plan across disjoint state categories to achieve alignment. In the supervised setting, CTVAE (Zhan et al., 2020) augments trajectory variational auto-encoders with trajectory style labels to perform imitation learning under style calibration, while BCPMI (Yang et al., 2025) performs a behavior cloning regression weighted by mutual information estimates between state-action pairs and style labels. Our method falls into the offline supervised learning category as in CTVAE and BCPMI as we employ supervised style labels to derive style reward signals for our policy to optimize. However, unlike both methods, we consider styles defined over subtrajectories. More specifically, in contrast to CTVAE, our method is model-free, and in contrast to BCPMI, it leverages RL signals to improve robustness to distribution shift and to jointly optimize task performance and style alignment.

**Goal-Conditioned RL.** Goal-Conditioned RL (GCRL) (Kaelbling, 1993; Liu et al., 2022; Park et al., 2025) encompasses methods that learn policies to achieve diverse goals efficiently and reliably. As our style alignment objective consists in visiting state-action pairs of high-probability to contribute to a given style, it shares with GCRL the same challenges of sparse rewards, long-term decision making and trajectory stitching. To address these challenges, Ghosh et al. (2021); Yang et al. (2022) combine imitation learning with Hindsight Experience Replay (HER) (Andrychowicz et al., 2017), while Chebotar et al. (2021); Kostrikov et al. (2022); Park et al. (2023); Canesse et al. (2025); Kobanda et al. (2025) additionally learn goal-conditioned value functions to extract policies using offline RL techniques. Unlike GCRL, which focuses on achieving specific goals, our framework addresses performing RL tasks under stylistic constraints. This can be viewed as a generalization from goal-reaching to executing diverse RL tasks while maintaining stylistic alignment.

## 3. Preliminaries

**Markov decision process.** We consider a $\gamma$-discounted Markov Decision Process (MDP) $\mathcal{M} = (\mathcal{S}, \mathcal{A}, \mu, p, r, \gamma)$, where $\mathcal{S}$ is the state space, $\mathcal{A}$ the action space, $\mu \in \Delta(\mathcal{S})$ the initial state distribution, $p : \mathcal{S} \times \mathcal{A} \rightarrow \Delta(\mathcal{S})$ the transition kernel, $r : \mathcal{S} \times \mathcal{A} \rightarrow [r_{\min}, r_{\max}]$ a reward function, and $\gamma \in [0, 1)$ the discount factor. An agent is modeled by a policy $\pi : \mathcal{S} \rightarrow \Delta(\mathcal{A})$. At timestep $t$, the agent observes $s_t \in \mathcal{S}$, selects $a_t \in \mathcal{A}$, and the environment transitions to $s_{t+1} \sim p(\cdot \mid s_t, a_t)$. This interaction generates a trajectory $\tau = (s_0, a_0, r_0, \ldots, s_{T-1}, a_{T-1}, r_{T-1}, s_T)$, where $r_t = r(s_t, a_t)$ and $T$ denotes a termination or truncation horizon. We assume access to a finite dataset $\mathcal{D}$ of such trajectories collected by an unknown behavior policy $\pi_{\mathcal{D}}$.

**Offline RL.** The RL objective is to learn a policy $\pi^{r,*} : \mathcal{S} \rightarrow \Delta(\mathcal{A})$ that maximizes the **task performance metric** $J(\pi)$, defined as the expected discounted cumulative sum of rewards, given any reward function $r$:

$$J(\pi) = \mathbb{E}_\pi \left[ \sum_{t=0}^\infty \gamma^t r(s_t, a_t) \right]. \tag{1}$$

In offline RL, this learning must be performed by only relying on a dataset of pre-recorded trajectories $\mathcal{D}$. Hence, regardless of the specific reward function $r$, offline RL algorithms must operate without further interaction with the environment. This makes it necessary to keep the learned policy close to the data-generating policy $\pi_{\mathcal{D}}$ in order to avoid out-of-distribution actions whose value estimates may be severely overestimated. This constraint directly conflicts with the objective of improving upon the behavior present in $\mathcal{D}$, creating a fundamental tension between policy improvement and distributional shift.

**Implicit Q-Learning.** To tackle this issue, the well-known IQL algorithm (Kostrikov et al., 2022) mitigates value overestimation by estimating the optimal value function through expectile regression:

$$\mathcal{L}_{V^r}(\phi^r) = \mathbb{E}_{(s,a)\sim D} \left[ \ell_\kappa^2 \left( Q_{\bar{\theta}^r}^r(s, a) - V_{\phi^r}^r(s) \right) \right] \tag{2}$$

$$\mathcal{L}_{Q^r}(\theta^r) = \mathbb{E}_{(s,a,s')\sim D} \left[ \left( r(s, a) + \gamma V_{\phi^r}^r(s') - Q_{\theta^r}^r(s, a) \right)^2 \right] \tag{3}$$

where $\ell_\kappa^2(u) = |\kappa - \mathbb{1}\{u < 0\}|u^2, \kappa \in [0.5, 1)$ is the expectile loss, an asymmetric squared loss that biases $V_{\phi^r}^r$ toward the upper tail of the $Q_{\bar{\theta}^r}^r$ distribution. The trained $V_{\phi^r}^r$ and $Q_{\theta^r}^r$ are then used to learn a policy network $\pi_{\psi^r}^r$ via Advantage-Weighted Regression (AWR) (Peng et al., 2019):

$$J_{\pi^r}(\psi^r) = \mathbb{E}_{(s,a)\sim\mathcal{D}} \left[ e^{\beta^r \cdot A_{\bar{\theta}^r,\phi^r}^r(s,a)} \log \pi_{\psi^r}^r(a \mid s) \right] \tag{4}$$

with $\beta \in (0, \infty]$ an inverse temperature and $A_{\bar{\theta}^r,\phi^r}^r(s, a) = Q_{\bar{\theta}^r}^r(s, a) - V_{\phi^r}^r(s)$ the advantage, which measures how

much better or worse action $a$ in state $s$ is compared to the baseline value. This procedure corresponds to cloning dataset state-action pairs with a bias toward actions with higher advantages.

## 4. Style-Conditioned Implicit Q-Learning

In this work, we aim to learn *stylized and high-performing policies* from a fixed offline dataset $\mathcal{D}$. In this section, we will present the designs of our method to tackle each of the challenges mentioned in the introduction: (1) Style definition, (2) Addressing state-style distribution shift and (3) Solving task and style misalignment.

### 4.1. Addressing Challenge 1: Style definition

**Style and diversity in imitation learning.** To train a policy toward a target behavior, traditional IL methods leverage $\mathcal{D}$ by mimicking its behaviors under the assumption of the combined expertise and homogeneity of its trajectories. In contrast, we assume that $\mathcal{D}$'s behaviors can possibly display a high amount of heterogeneity. Previous literature (Zhan et al., 2020; Mao et al., 2024; Yang et al., 2025) describes this heterogeneity through various definitions of behavior styles. Denoting $\tilde{\mathcal{T}}$ as the set of (overlapping) subtrajectories, we can generalize those definitions by defining a style as the **labeling** of a subtrajectory $\tau_{t:t+h} \in \tilde{\mathcal{T}}$ given a comparison **criterion** with respect to a **task** to perform. Hence, a style translates into a specific way to carry out a given task under a given criterion. A **task** in the MDP framework is generally defined through its reward function $r : \mathcal{S} \times \mathcal{A} \to [r_{\min}, r_{\max}]$ to maximize along the trajectory. Given a task, an agent can display a range of behaviors that varies greatly. A **criterion** $\lambda : \tilde{\mathcal{T}} \to \mathcal{L}(\lambda)$ is a tool to describe such variations. It can range from "*the vector of an unsupervised learned trajectory encoder*" to "*the speed class of my agent*" and projects any subtrajectory into a **label** in $\mathcal{L}(\lambda)$. For instance, we can have $z \in \mathcal{L}(\lambda) = \mathbb{R}^d$ or "*fast*" $\in \mathcal{L}(\lambda) = \{$"*slow*", "*fast*"$\}$. Consequently, in the more general sense, aligning one's behavior to a given **behavior style** corresponds to generating subtrajectories that satisfy a certain label, given a criterion and a task.

**Style labeling and data programming.** The various definitions of behavior styles in the literature can be divided into unsupervised settings (Li et al., 2017; Hausman et al., 2017; Wang et al., 2017; Mao et al., 2024; Petitbois et al., 2025) and supervised settings (Zhan et al., 2020; Yang et al., 2025). In particular, following Zhan et al. (2020); Yang et al. (2025), we focus on the data programming (Ratner et al., 2016) paradigm, using labeling functions as the criterion. However, unlike in Zhan et al. (2020); Yang et al. (2025), where labeling functions are defined on full trajectories given any criterion $\lambda$, we define ours as hard-coded functions on subtrajectories $\lambda : \tilde{\mathcal{T}} \to [\![0, |\lambda| - 1]\!]$, with $|\lambda|$

the number of categories of $\lambda$. Using such labeling functions has several benefits. As noted in Zhan et al. (2020), labeling functions are simple to specify yet highly flexible. They reduce **labeling cost** by eliminating manual annotation, which is often time-consuming and expensive, and, crucially, they enhance interpretability, a key limitation of unsupervised approaches, thereby enabling clearer notions of **interpretability** and more direct **alignment measurement**. While previous works such as Zhan et al. (2020); Yang et al. (2025) have focused on trajectory-level labels $\lambda(\tau)$, we argue that relying on per-timestep labeling functions, defined in our framework as labels of windows, is a more pragmatic choice. Indeed, as different styles can operate at different timescales, styles can in fact vary across a trajectory, which can lead to avoidable **credit assignment** issues. As such, given a labeling function $\lambda$, we annotate the dataset $\mathcal{D}$ by marking each state-action pair $(s_t, a_t)$ of each of its trajectories $\tau$ as "contributing" to the style of its corresponding window of radius $w(\lambda)$:

$$\lambda(\mathcal{D}) = \{(s_t, a_t, s_{t+1}, z_t^\lambda) : t = 0, \ldots, T - 1, \ \tau \in \mathcal{D}\},$$

with $z_t^\lambda = \lambda(\tau_{t-w(\lambda)+1:t+w(\lambda)})$. Boundary handling (e.g., truncated windows) is implementation-specific and omitted for clarity. This per-timestep annotation allows styles to vary along trajectories and mitigates credit assignment issues arising for episode-scale labeling as in Yang et al. (2025).

**True style alignment objective.** Consequently, given a criterion $\lambda$ and a target style $z \in \mathcal{L}(\lambda)$, a natural objective for style alignment would correspond to the generation of trajectories which maximally exhibit style $z$:

$$S^{\mathbb{1}}(\pi, \lambda, z) = \mathbb{E}_\pi \left[ \sum_{t=0}^\infty \gamma^t \mathbb{1}\{z_t^\lambda = z\} \right]. \tag{5}$$

This objective is well-defined but generally *non-Markovian* with respect to the original state space, since the label depends on a trajectory window (past and future). Optimizing (5) directly would require augmenting the state with sufficient trajectory context. Nevertheless, a simpler probabilistic relaxation can be obtained by conditioning on the current state-action pair:

$$p_\pi^\lambda(z \mid s_t, a_t) = \mathbb{P}_\pi\left(z_t^\lambda = z \mid s_t, a_t\right), \tag{6}$$

yielding the objective

$$\tilde{S}^p(\pi, \lambda, z) = \mathbb{E}_\pi \left[ \sum_{t=0}^\infty \gamma^t p_\pi^\lambda(z \mid s_t, a_t) \right]. \tag{7}$$

However, computing $p_\pi^\lambda(z \mid s, a)$ for arbitrary policies $\pi$ requires modeling long-horizon trajectory distributions and is generally intractable. This is especially the case when optimizing the policy $\pi$, making it change along the training.

**Surrogate Markov style objective.** Instead, in order to fit the MDP and RL framework, we construct a *surrogate Markovian style reward* from the demonstration dataset. Using the annotated dataset $\lambda(\mathcal{D})$, we estimate a label predictor $p_{\pi_{\mathcal{D}}}^\lambda(z \mid s, a)$, representing the probability that a state-action pair $(s, a)$ lies in the center of a window labeled $z$ in the dataset $\lambda(\mathcal{D})$ generated by $\pi_{\mathcal{D}}$. We then define the surrogate per-label **Style-Conditioned RL (SCRL)** objective:

$$S^p(\pi, \lambda, z) = \mathbb{E}_\pi \left[ \sum_{t=0}^\infty \gamma^t p_{\pi_{\mathcal{D}}}^\lambda(z \mid s_t, a_t) \right]. \quad (8)$$

This defines a standard Markov reward function $r_{\mathcal{D}}^\lambda(s, a, z) = p_{\pi_{\mathcal{D}}}^\lambda(z \mid s, a)$, and therefore a well-posed MDP optimization problem. In the following, we will refer to this surrogate objective when writing about style alignment optimization. In general, it provides a tractable approximation that encourages policies to visit state-action pairs judged as style-consistent by the dataset-derived label model. This casting of (7) to (8) enables the direct application of standard offline reinforcement learning methods on the demonstration dataset $\mathcal{D}$. We show experimentally that this objective still permits high alignment when computing the empirical true style alignment metric in Equation 16.

## 4.2. Addressing Challenge 2: State-style distribution shift

The surrogate style reward in Equation (8) is typically sparse: many styles are only exhibited in a small subset of states, and some state–style pairs $(s, z)$ may be entirely absent from the labeled dataset $\lambda(\mathcal{D})$. As a consequence, direct behavior cloning of style-conditioned policies is often impossible, since no demonstration exists for many relevant $(s, a, z)$ tuples. Figure 1 illustrates this issue: transitions $(s, a)$ that do not appear in trajectories labeled with style $z$ provide no immediate supervision for aligning with $z$, calling for planning capabilities beyond single-step imitation. Moreover, a target style $z$ may never appear within a single complete trajectory in $\mathcal{D}$, requiring the policy to *stitch* together partial behaviors from different trajectories. This further emphasizes the need for long-term decision-making and trajectory stitching in order to achieve consistent style alignment.

To optimize for style alignment, we introduce SCIQL, a simple adaptation of IQL which employs the same principles of relabeling as the GCRL literature (Park et al., 2025) to learn an optimal policy $\pi^{\lambda,*} : \mathcal{S} \times \mathcal{L}(\lambda) \to \Delta(\mathcal{A})$ for any given criterion $\lambda$:

$$\forall z \in \mathcal{L}(\lambda), \pi^{\lambda,*}(\cdot \mid \cdot, z) \in \Pi^\lambda(z) = \arg\max_\pi S^p(\pi, \lambda, z) \quad (9)$$

where $\Pi^\lambda(z)$ denotes the set of policies achieving maximal alignment with style $z$. This objective corresponds to the

per-criterion **SCRL** objective. As in IQL, SCIQL first fits the optimal style-conditioned value functions through neural networks $V_{\phi^\lambda}^\lambda$ and $Q_{\theta^\lambda}^\lambda$ using expectile regression:

$$\mathcal{L}_{V^\lambda}(\phi^\lambda) = \mathbb{E}_{\substack{(s,a)\sim\mathcal{D} \\ z\sim\lambda(\mathcal{D})}} \left[ \ell_\kappa^2 (Q_{\theta^\lambda}^\lambda(s, a, z) - V_{\phi^\lambda}^\lambda(s, z)) \right], \quad (10)$$

$$\mathcal{L}_{Q^\lambda}(\theta^\lambda) = \mathbb{E}_{\substack{(s,a,s')\sim\mathcal{D} \\ z\sim\lambda(\mathcal{D})}} \left[ \left( \chi_{\omega^\lambda}^\lambda(s, a, z) + \gamma V_{\phi^\lambda}^\lambda(s', z) \right.\right. \quad (11)$$
$$\left.\left. - Q_{\theta^\lambda}^\lambda(s, a, z) \right)^2 \right].$$

with $\chi_{\omega^\lambda}^\lambda(s, a, z)$ an estimator of $p_{\pi_{\mathcal{D}}}^\lambda(z \mid s, a)$. Comparing several strategies, we empirically found (see Appendix E.1) that taking $\chi_{\omega^\lambda}^\lambda(s, a, z) = \mathbb{1}\{z = z_t^\lambda\}$ with $z_t^\lambda$ the associated label of $(s, a, s')$ within $\lambda(\mathcal{D})$ to be one of the best performing methods, which we kept for its simplicity. This sampling of style labels outside the joint distribution $p^{\lambda(\mathcal{D})}(s, a, z)$ addresses **distribution shift**. Indeed, after that, we extract a style-conditioned policy $\pi_{\psi^\lambda}^\lambda$ through AWR by optimizing:

$$J_{\pi^\lambda}(\psi^\lambda) = \mathbb{E}_{\substack{(s,a)\sim\mathcal{D} \\ z\sim\lambda(\mathcal{D})}} \left[ e^{\beta^\lambda \cdot A_{\theta^\lambda,\phi^\lambda}^\lambda(s,a,z)} \log \pi_{\psi^\lambda}^\lambda(a \mid s, z) \right] \quad (12)$$

with $A_{\theta^\lambda,\phi^\lambda}^\lambda(s, a, z) = Q_{\theta^\lambda}^\lambda(s, a, z) - V_{\phi^\lambda}^\lambda(s, z)$ the advantage. This objective drives $\pi_{\psi^\lambda}^\lambda$ to copy the dataset's actions with a bias toward actions likely to lead in the future to the visitation of state-action pairs of high likelihood of contribution to the conditioning style. This formulation effectively works with styles outside of the joint distribution and leads, as shown in section 5.2, to a more **robust style alignment**.

## 4.3. Addressing Challenge 3: Task optimization under style alignment

As the behaviors in $\mathcal{D}$ may vary in quality, our conceptual objective is to select, among these style-optimal policies, one that attains the highest task performance:

$$\forall z \in \mathcal{L}(\lambda), \pi^{r|\lambda,*}(\cdot \mid \cdot, z) \in \arg\max_{\pi \in \Pi^\lambda(z)} J(\pi). \quad (13)$$

This formulation expresses our two goals. First, we want to learn a style-conditioned policy $\pi^{\lambda,*}$ such that $\forall z \in \mathcal{L}(\lambda), \pi^{\lambda,*}(\cdot \mid \cdot, z) \in \Pi^\lambda(z)$. Secondly, we want to learn $\pi^{r|\lambda,*}$ by increasing the task performance of $\pi^{\lambda,*}$ without hurting its style alignment. The objective in Equation (13) involves projecting onto the set of style-optimal policies $\Pi^\lambda(z)$ before optimizing task performance. Explicitly computing this set is infeasible in practice: it would require solving a full offline RL problem to global optimality for the style objective, enumerating all optimal policies, and then performing a second constrained optimization step within

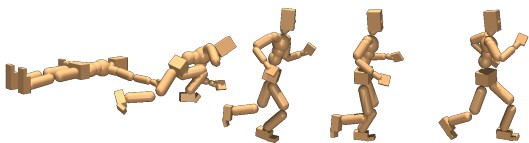

*Figure 1.* **Long-term decision-making and stitching challenges for style alignment optimization.** Achieving movement styles such as high-speed running may require standing up and accelerating, which means navigating through different speed styles and demands **long-term decision making**. Also, trajectories in $\mathcal{D}$ may not cover all the speed styles, calling for **trajectory stitching** such as (slow → medium) and (medium → fast).

this set. Furthermore, approximation errors in value estimation and limited dataset coverage make exact constraint enforcement brittle in the offline setting. These difficulties call for a practical policy improvement rule that approximates the projection behavior of Equation (13) while remaining stable under offline training.

For each target label $z$, we seek to maximize task performance *within* the set of policies that maximize style alignment. A natural way to formalize this objective is through multi-objective or constrained reinforcement learning, for instance by optimizing a weighted combination of task and style returns or by maximizing task performance subject to a constraint on style alignment. While these frameworks are well-suited when a precise trade-off or constraint threshold is available, in our setting such a threshold is not naturally defined: different styles may admit different achievable alignment scores, and selecting a universal constraint level introduces an additional hyperparameter that is difficult to tune from data alone. Moreover, in the offline setting, constrained optimization approaches typically introduce additional dual variables or constraint hyperparameters whose optimization can be sensitive to value approximation errors and limited dataset coverage, making stable training challenging in practice. Nevertheless, exploring principled constrained or multi-objective formulations for stylized offline reinforcement learning is an interesting direction for future work.

Instead, we propose to adopt a more direct *policy improvement principle* that approximates the projection behavior of Equation (13) while remaining simple and stable under offline training. Let $\pi^\lambda$ denote a style-aligned policy for label $z$, with associated style advantage $A^\lambda(s, a, z)$, and let $A^r(s, a)$ denote the task advantage. Positive style advantage indicates transitions that improve alignment with the target style, while negative values correspond to detrimental transitions. Our objective is to incorporate task improvement signals primarily when they are unlikely to harm style alignment, thereby encouraging increases in task performance without sacrificing controllability.

**Gated Advantage Weighted Regression (GAWR).** We implement this principle through a gated combination of style and task advantages. Concretely, we define the *gated advantage*

$$\xi(A^\lambda, A^r)(s, a, z) = A^\lambda(s, a, z) + \sigma\big(A^\lambda(s, a, z)\big) \cdot A^r(s, a),$$
(14)

where $\sigma(\cdot)$ is the sigmoid function. Advantages can be normalized with an exponential moving average in order to have a similar order of magnitude.

We then perform a weighted behavior cloning update using $\xi$ as the preference score, yielding the style-conditioned task policy $\pi^{r|\lambda}$:

$$J_{\pi^{r|\lambda}}(\psi^{r|\lambda}) = \mathbb{E}_{\substack{(s,a) \sim \mathcal{D} \\ z \sim \lambda(\mathcal{D})}} \Big[ e^{\beta^{r|\lambda} \xi\big(A^\lambda_{\bar\theta^\lambda, \phi^\lambda}, A^r_{\bar\theta^r, \phi^r}\big)(s,a,z)} \times$$
$$\log \pi^{r|\lambda}_{\psi^{r|\lambda}}(a \mid s, z) \Big].$$
(15)

Compared to an unconstrained combination such as $A^\lambda(s, a, z) + A^r(s, a)$, which may allow task optimization to dominate and reduce style alignment, GAWR explicitly modulates the task signal based on the style signal. This yields a practical approximation to the projection in Equation (13) and empirically produces improvements in task performance while preserving high style alignment.

We provide the full training pipeline in Algorithm 1. As in IQL and related methods, while written separately, (Kostrikov et al., 2022; Park et al., 2023), value learning and policy extraction can be performed jointly in practice. This allows the entire training procedure to be performed within a single loop, considerably simplifying the process.

---

**Algorithm 1** SCIQL with GAWR

**Input:** offline dataset $\mathcal{D}$, labeling function $\lambda$
Initialize $\phi^r, \phi^\lambda, \theta^r, \bar\theta^r, \theta^\lambda, \bar\theta^\lambda, \psi^{r|\lambda}$
  # Train the task value functions
**while** not converged **do**
    $\phi^r \leftarrow \phi^r - \nu_{V^r} \nabla \mathcal{L}_{V^r}(\phi^r)$ (Equation 2)
    $\theta^r \leftarrow \theta^r - \nu_{Q^r} \nabla \mathcal{L}_{Q^r}(\theta^r)$ (Equation 3)
    $\bar\theta^r \leftarrow (1 - \upsilon_{\text{Polyak}})\bar\theta^r + \upsilon_{\text{Polyak}}\theta^r$
**end while**
  # Train the style value functions
**while** not converged **do**
    $\phi^\lambda \leftarrow \phi^\lambda - \nu_{V^\lambda} \nabla \mathcal{L}_{V^\lambda}(\phi^\lambda)$ (Equation 10)
    $\theta^\lambda \leftarrow \theta^\lambda - \nu_{Q^\lambda} \nabla \mathcal{L}_{Q^\lambda}(\theta^\lambda)$ (Equation 11)
    $\bar\theta^\lambda \leftarrow (1 - \upsilon_{\text{Polyak}})\bar\theta^\lambda + \upsilon_{\text{Polyak}}\theta^\lambda$
**end while**
  # Train the policy $\psi^{r|\lambda}$ through GAWR
**while** not converged **do**
    $\psi^{r|\lambda} \leftarrow \psi^{r|\lambda} + \nu_{\pi^{r|\lambda}} \nabla J_{\pi^{r|\lambda}}(\psi^{r|\lambda})$ (Equation 15)
**end while**
**return** Policy $\pi^{r|\lambda}_{\psi^{r|\lambda}}$

---

# 5. Experiments

## 5.1. Experimental setup

After introducing environments in section 5.1.1, we tackle the following experimental questions:

1. How does SCIQL compare to previous work on style alignment?

2. Does GAWR help SCIQL perform style-conditioned task-performance optimization?

3. How does SCIQL compare to previous work on style-conditioned task-performance optimization?

### 5.1.1. ENVIRONMENTS, TASKS, LABELS AND DATASETS

**Circle2d** is a modified version of the environment from Li et al. (2017) and consists of a 2D plane where an agent can roam within a confined square to draw a target circle. For this environment, we define the labels: `position`, `movement_direction`, `turn_direction`, `radius`, `speed`, and `curvature_noise`. We generate two datasets using a hard-coded agent that draws circles with various centers and radii, orientations (clockwise and counter-clockwise), speeds, and action noise levels. The first dataset, `circle2d-inplace-v0`, is obtained by drawing the circle directly from the start position, while the `circle2d-navigate-v0` dataset is obtained by navigating to a target position before drawing the circle. **HalfCheetah** (Todorov et al., 2012) is a task where the objective is to control a planar 6-DoF robot to move as far as possible in the forward direction. For this environment, we define the labels: `speed`, `angle`, `torso_height`, `backfoot_height`, and `frontfoot_height`. We train a diverse set of HalfCheetah policies using SAC (Haarnoja et al., 2018) to generate three datasets: `halfcheetah-fix-v0`, where the policy is fixed throughout the trajectory, `halfcheetah-stitch-v0`, where trajectories are split into short segments, and `halfcheetah-vary-v0`, where the policy changes during the trajectory. **HumEnv** (Tirinzoni et al., 2025) is a higher dimensional task consisting in controlling a SMPL skeleton (Loper et al., 2015) with 358-dimensional observations through a 69-dimensional action space to move as fast as possible on a flat plane. In `humenv-simple-v0`, the humanoid is initialized in a standing position. We generate a stylized dataset using the Metamotivo-M1 model provided in Tirinzoni et al. (2025), leading to various ways of moving at different heights and speeds. We focus on a `simple-head_height` criterion of 2 labels, `low` and `high`. In `humenv-complex-v0`, the humanoid is initialized in a lying down position, and the dataset is generated as in `humenv-simple-v0`, but with style variations within the trajectory. Also, in `humenv-complex-v0`, we define a

complex-speed criterion of 3 labels: `immobile`, `slow` and `fast`, and a finer `complex-head_height` criterion of 3 labels: `low`, `medium` and `high`. Further details about each environment, task, labeling function and dataset are provided in Appendix A.

### 5.1.2. BASELINES AND MODEL DETAILS

We compare SCIQL against external state-of-the-art algorithms and a hierarchy of ablations designed to isolate the contributions of SCIQL's components. For the ablations, we begin with standard **BC** (Pomerleau, 1991) as a non-conditioned reference. We then introduce **Conditioned BC (CBC)**, which incorporates style conditioning using the current trajectory style. Finally, to analyze the benefits of style relabeling, we introduce **SCBC**, an IL variant of **SCIQL** which performs hindsight style relabeling by sampling style labels from the future trajectory, but without value functions. For external comparisons, we evaluate against **BCPMI** (Yang et al., 2025), which extends CBC via mutual-information weighting, and an adapted version of **SORL** (Mao et al., 2024) (see Appendix C), which serves as the primary benchmark for optimizing task performance under style constraints. Further details on architectures and hyperparameters are provided in Appendix B and C.

## 5.2. Results on style alignment

Our first set of experiments evaluates the capability of SCIQL to achieve style alignment compared to baselines. For each style label $z \in \mathcal{L}(\lambda)$ of each criterion $\lambda$, we perform 10 rollouts across 5 seeds, conditioned on $z$ (except BC, which does not support label conditioning). Each generated trajectory $\tau = \{(s_t, a_t), t \in \{0, \ldots, |\tau| - 1\}\}$ is then annotated as $\lambda(\tau) = \{(s_t, a_t, z_t^\lambda), t \in \{0, \ldots, |\tau| - 1\}\}$ with $z_t^\lambda = \lambda(\tau_{t-w(\lambda)+1:t+w(\lambda)}), \forall t \in \{0, \ldots, |\tau| - 1\}$. For each annotated trajectory, we compute its empirical normalized undiscounted style alignment:

$$\hat{S}^{\mathbb{1}}(\lambda(\tau), z) = \frac{1}{|\tau|} \sum_{t=0}^{|\tau|-1} \mathbb{1}\{z_t^\lambda = z\}, \qquad (16)$$

where the normalization by the trajectory length $|\tau|$ ensures that $\hat{S}^{\mathbb{1}}(\lambda(\tau), z) \in [0, 1]$, which hence represents the fraction of timesteps labeled as contributing to the target label. We then average alignments over 10 episodes to compute the empirical normalized undiscounted style alignment of our policy, $\hat{S}^{\mathbb{1}}(\pi, \lambda, z)$, which can be seen as the analogue of a GCRL success rate in the SCRL context. Because of the multiplicity of criteria and labels (see Appendix D), we report average alignments across all criteria and labels in Table 1, with full results provided in Appendix D. Standard deviations are computed as the average across 5 seeds for the different tested $(\lambda, z)$. We observe that SCIQL achieves the best style alignment performance by a large margin

*Table 1.* **Style alignment results**. SCIQL achieves the best style alignment performance by a large margin compared to baselines for every dataset. The results are the averages over the criteria, labels and 5 seeds. A more detailed table can be found in Appendix D.

| Dataset | BC | CBC | BCPMI | SORL ($\beta = 0$) | SCBC | SCIQL |
|---|---|---|---|---|---|---|
| `circle2d-inplace-v0` | $29.1_{\pm6.3}$ | $58.6_{\pm2.3}$ | $58.9_{\pm2.6}$ | $58.9_{\pm2.7}$ | $68.6_{\pm2.0}$ | $\mathbf{74.6}_{\pm9.3}$ |
| `circle2d-navigate-v0` | $29.1_{\pm5.3}$ | $58.9_{\pm2.7}$ | $59.9_{\pm2.3}$ | $60.0_{\pm3.3}$ | $67.2_{\pm1.8}$ | $\mathbf{75.5}_{\pm4.7}$ |
| `halfcheetah-fix-v0` | $30.0_{\pm5.9}$ | $51.2_{\pm9.0}$ | $58.1_{\pm8.4}$ | $53.1_{\pm10.6}$ | $58.0_{\pm5.3}$ | $\mathbf{78.0}_{\pm1.8}$ |
| `halfcheetah-stitch-v0` | $30.0_{\pm6.8}$ | $52.1_{\pm7.6}$ | $58.9_{\pm11.3}$ | $48.4_{\pm12.5}$ | $57.4_{\pm4.7}$ | $\mathbf{78.0}_{\pm1.1}$ |
| `halfcheetah-vary-v0` | $30.0_{\pm4.5}$ | $52.0_{\pm12.0}$ | $52.6_{\pm17.2}$ | $46.7_{\pm9.5}$ | $31.7_{\pm4.2}$ | $\mathbf{78.9}_{\pm0.7}$ |
| `humenv-simple-v0` | $50.0_{\pm44.4}$ | $89.1_{\pm22.0}$ | $79.2_{\pm26.7}$ | $79.4_{\pm26.9}$ | $99.6_{\pm0.0}$ | $\mathbf{99.6}_{\pm0.0}$ |
| `humenv-complex-v0` | $33.3_{\pm4.0}$ | $47.1_{\pm12.8}$ | $44.6_{\pm18.4}$ | $47.7_{\pm6.9}$ | $33.2_{\pm3.5}$ | $\mathbf{83.5}_{\pm6.2}$ |

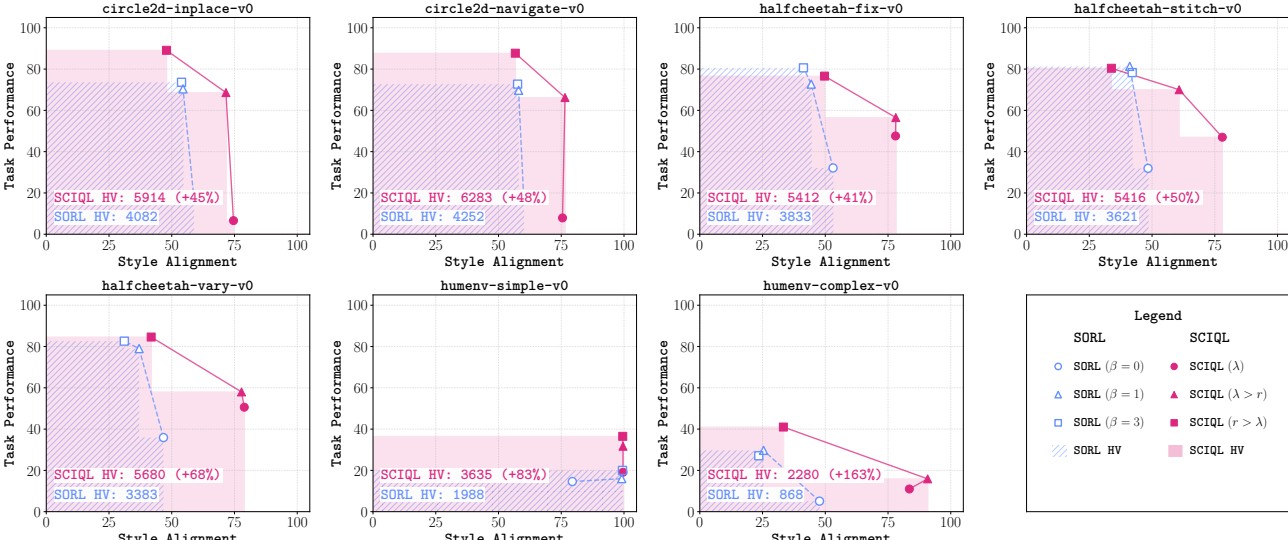

*Figure 2.* **Pareto fronts and hypervolumes of SORL and SCIQL.** We compare **SORL** (in dashed blue) and **SCIQL** (in full red). The shaded areas (▨, ▨) represent the hypervolumes covered by the methods. Markers indicate different trade-off configurations: **SORL** is evaluated at $\beta = 0$ (○), $\beta = 1$ (△), and $\beta = 3$ (□). **SCIQL** is evaluated with style only $\lambda$ (●), style-prioritized $\lambda > r$ (▲), and task-prioritized $r > \lambda$ (■). SCIQL consistently achieves a larger hypervolume and dominates the Pareto frontier.

compared to baselines for every dataset, highlighting its effectiveness in long-term decision making and stitching, unlike prior methods. In particular, the performance gap between BC and CBC underscores the necessity of style conditioning. Moreover, the similar performance of SORL in imitation mode ($\beta = 0$), BCPMI, and CBC can be explained by the similarity of their objectives (see Appendix C), all corresponding to a weighted CBC without style relabeling. The performance gap between SCBC and the other baselines further highlights the importance of integrating trajectory stitching and style relabeling within stylized policies, while the dominance of SCIQL demonstrates the additional benefits of value learning, which augments relabeling by integrating randomly sampled styles during training and enables more effective policy extraction overall.

### 5.3. Results on style-conditioned task-performance optimization

To evaluate the capability of SCIQL to perform style-conditioned task-performance optimization, we plot in Fig-

ure 2 the average style alignments and normalized returns of SCIQL without GAWR ($\lambda$), with a style-based GAWR ($\lambda > r$), and with a reward-based GAWR ($r > \lambda$) to compare to a task-performance prioritized setting. We compare against SORL with various temperatures $\beta$, which control the importance of task performance in the SORL objective (see Appendix C).

**Task-performance optimization under style alignment.** First, we observe that incorporating task reward signals with GAWR increases the score of SCIQL while preserving the style alignment in nearly all environments, showing a decrease only for `halfcheetah-stitch-v0`. Additionally, SCIQL ($r > \lambda$), which prioritizes task performance over style alignment, achieves higher task performance, as expected, but at the cost of style alignment. **This shows that the GAWR effectively enables the increase of task performance of our policies while maintaining style alignment.**

**Achieving overall better joint performance.** Secondly, we compute the hypervolumes (HV) of SORL and SCIQL variants and observe that SCIQL achieves a substantial im-

provement of +41% to +163% (see Figure 2) over SORL across environments. This indicates that SCIQL achieves a better overall task-performance to style-alignment trade-off than SORL. In particular, $SCIQL(\lambda > r)$ lies closer to the ideal point $(100, 100)$, corresponding to a reduction in Euclidean distance to the ideal point of 18-28%. **This shows that SCIQL reaches a stronger joint performance between objectives than SORL, effectively shifting the Pareto frontier closer to theoretical optimality**. See Appendix D and Appendix E for more details and ablations.

## 6. Conclusion

We propose a novel general definition of behavior styles within the sequential decision making framework and instantiate it using labeling functions to define **interpretable** styles with a low **labeling cost** and easy **alignment measurement** while effectively avoiding unnecessary **credit assignment** issues by relying on subtrajectory labeling. We then present the SCIQL algorithm which leverages Gated AWR to solve long-term decision making and trajectory stitching challenges while providing superior performance in both style alignment and style-conditioned task performance compared to previous work.

Our framework suggests several promising directions for future work. An interesting next step would be to find ways to scale it to a multiplicity of criteria. Finding mechanisms to enhance the representation span of labeling functions could also be interesting, as well as integrating zero-shot capabilities to generate on-the-fly style-conditioned reinforcement learning policies. Additionally, incorporating state-level similarity measures into our framework offers a promising direction for grounding style labels more directly in intrinsic policy behavior, instead of extrinsic temporal windows, thus improving the explainability of our policies.

## Acknowledgments

We would like to thank Ludovic Denoyer for helping initiate this project and for contributing to the early discussions that shaped its development. This work was granted access to the HPC resources of IDRIS under the allocations AD011014679R1, AD011014679R2 and A0181016109 made by GENCI.

## Impact Statement

We acknowledge that our framework might introduce concrete misuse risks. Style labels may encode annotator, cultural, or demographic biases, leading policies to reproduce or amplify discriminatory behaviors. It could also enable behavioral forgery, allowing agents to imitate the style of specific individuals, groups, or institutions for deceptive purposes. In addition, inferred style labels may reveal sensitive information about users, including preferences, habits, abilities, or demographic traits. These risks call for explicit safeguards, including careful label design, bias audits, restrictions on identity-level imitation and consent requirements for user-specific style modeling.

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

# A. Environments, tasks, labels and datasets

In this section, we detail our environments, tasks, labels and datasets.

## A.1. Circle2d

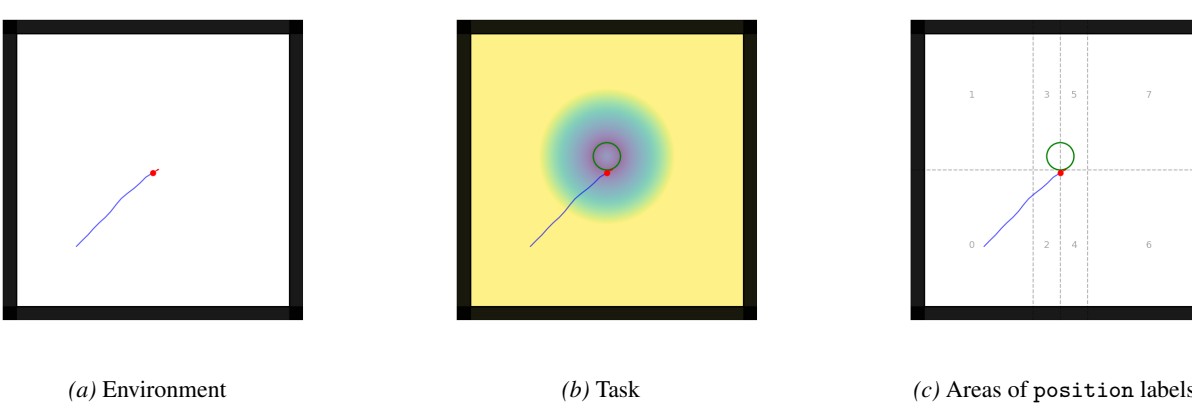

*(a)* Environment    *(b)* Task    *(c)* Areas of `position` labels

*Figure 3.* **Circle2d environment visualizations.**

**Environment**    The **Circle2d** environment consists of a 2D plane where an agent can roam around within a confined square. Its state space $\mathcal{S}$ corresponds to the history of the previous 4 $(x_{\mathrm{agent}}, y_{\mathrm{agent}}, \theta_{\mathrm{agent}}) \in [x_{\min}, x_{\max}] \times [y_{\min}, y_{\max}] \times [\theta_{\min}, \theta_{\max}] = [-50.0, 50.0] \times [-50.0, 50.0] \times [-\pi, \pi]$, padded if needed by repeating the oldest triplet (namely for the beginning of the trajectory). Its action space $\mathcal{A}$ is $[-1, 1]^2$ where the first dimension maps onto an angular shift $\Delta\theta \in [\Delta\theta_{\min}, \Delta\theta_{\max}] = [-\pi, \pi]$ in radians and the second dimension maps onto a speed in $[v_{\min}, v_{\max}] = [0.5, 3.0]$. At first, the environment is initialized by sampling a random position from $[0.7 \cdot x_{\min}, 0.7 \cdot x_{\max}] \times [0.7 \cdot y_{\min}, 0.7 \cdot y_{\max}]$ and a random orientation from $[-\pi, \pi]$. At each timestep $t$, given a state $s_t$ and an action $a_t$, the agent rotates by the corresponding $\Delta\theta_t$ before moving by the displacement vector (related to speed) $v_t$. The episode is truncated after 1000 timesteps have been reached. We display a minimal visual example of our environment in Figure 3a.

**Task**    In **Circle2d**, we define the task as drawing a target circle $\mathcal{C}$ given its center's xy position $p_{\mathcal{C}}$ and its radius $\rho_{\mathcal{C}}$ and encode it by a reward: $r(s_t, a_t) = -|\|p_t - p_{\mathcal{C}}\| - \rho_{\mathcal{C}}|$ where $p_t$ is the agent's xy position at timestep $t$. In this work, we consider the same fixed target circle $\mathcal{C}$ along experiments and we display its associated reward colormap in Figure 3b.

**Datasets**    We generate for this environment two datasets by using a hard-coded agent which draws circles of various centers and radii, with different orientations (clockwise and counter-clockwise) and different speed and noise levels on the actions. The first dataset `circle2d-inplace-v0` is obtained by directly performing the circle at start position, while the `circle2d-navigate-v0` dataset is obtained by navigating to a target position before drawing the circle. We plot in Figure 4 the datasets' trajectories.

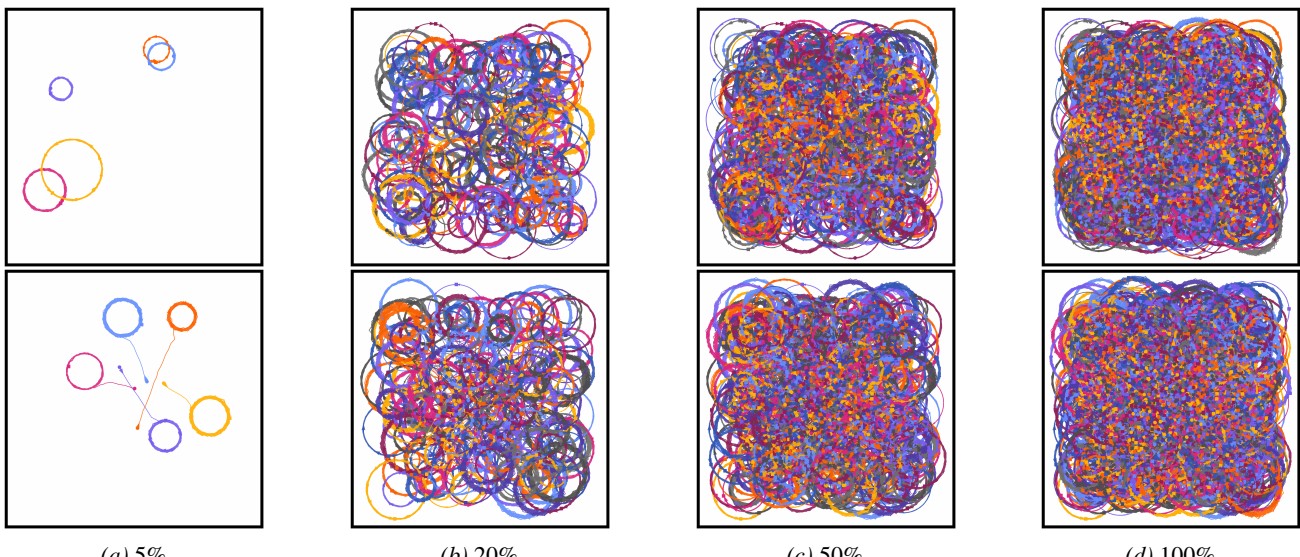

*(a) 5%*          *(b) 20%*          *(c) 50%*          *(d) 100%*

*Figure 4.* **Circle2d datasets trajectory visualizations at different percentages.** The top row corresponds to the `circle2d-inplace-v0` while the bottom row corresponds to the `circle2d-navigate-v0`.

**Criteria and labels**    We present below the various labeling functions we designed for **Circle2d**.

• `position`: The position labeling function $\lambda_{\text{position}}$ partitions the 2D plane into a fixed grid and assigns to each timestep the index of the cell containing the current position. Concretely, the $x$-axis range $[-30, 30]$ (real units) is split uniformly into 4 bins and the $y$-axis is split at 0 into 2 bins, yielding $4 \times 2 = 8$ areas. At timestep $t$, with window radius $w(\lambda_{\text{position}})$, we read every $(x_{t'}, y_{t'})$ in the window $\tau_{t-w(\lambda_{\text{position}})+1:t+w(\lambda_{\text{position}})}$ and set the label as the majority area. The label set is $\mathcal{L}(\lambda_{\text{position}}) = [\![0, 7]\!]$. In practice, we take $w(\lambda_{\text{position}}) = 1$ to mitigate unnecessary credit assignment issues. We plot in Figure 5 the corresponding visuals and histograms.

• `movement_direction`: The movement-direction labeling function $\lambda_{\text{md}}$ discretizes the instantaneous displacement direction. For each timestep $t'$, we compute the displacement vector $\Delta p_{t'} = p_{t'+1} - p_{t'}$ and its corresponding angle $\theta_{t'} = \text{atan2}(\Delta y_{t'}, \Delta x_{t'})$. We then uniformly quantize $[-\pi, \pi)$ into $K = 8$ bins. If $\|\Delta p_{t'}\| < 0.1$ (real units) for a frame, it contributes an undetermined class $u$ (non-promptable). With window radius $w(\lambda_{\text{md}})$, the label at $t$ is the majority direction category over $\tau_{t-w(\lambda_{\text{md}})+1:t+w(\lambda_{\text{md}})}$. Thus $\mathcal{L}(\lambda_{\text{md}}) = [\![0, 8]\!]$, with promptable bins 0..7 and $u = 8$. In practice we use $w(\lambda_{\text{md}}) = 1$ to mitigate unnecessary credit assignment issues. See Figure 6 for visuals and histograms.

• `turn_direction`: The turn-direction labeling function $\lambda_{\text{turn}}$ inherently operates on a centered temporal window to estimate local angular velocity. Let $(\theta_t)_t$ be the unwrapped heading. On a window $\tilde{\tau}_t$ of odd size $W(\lambda_{\text{turn}})$ (default size 11) centered at $t$, for each timestep $t'$, we form $\Delta\theta_{t'} = \theta_{t'+1} - \theta_{t'}$ and compute $\bar{\omega}_t$ the average $(\Delta\theta_{t'})_{t'}$ of $\tilde{\tau}_t$. If $|\bar{\omega}_t| < 0.1$ rad/step we label "straight," else "left" if $\bar{\omega}_t > 0$ (counter-clockwise) and "right" if $\bar{\omega}_t < 0$ (clockwise). We set $\mathcal{L}(\lambda_{\text{turn}}) = \{0, 1, 2\}$ with `right` $= 0$, `left` $= 1$, `straight` $= 2$ (non-promptable). We plot in Figure 7 its visuals and histograms.

• `radius_category`: The radius labeling function $\lambda_{\text{radius}}$ also works directly on centered windows. First, on a short window of size $W_{\text{radius}}^{\text{straight}}$ (default size 11) centered at $t$, we test straightness via the mean absolute heading increment, if it is below $0.1$ rad/step, the label is "straight." Otherwise, on a larger window of size $W_{\text{radius}}^{\text{circular}}$ (default size 51) centered at $t$, we fit a circle by least squares and take its radius $\text{radius}_t$. We uniformly partition $[2, 11]$ (real units) into $K = 3$ bins and assign the corresponding bin, the straight case is encoded as bin $K$. Thus $\mathcal{L}(\lambda_{\text{radius}}) = [\![0, K]\!]$, where $0..K-1$ denote increasing-radius curved motion and $K$ denotes straight (non-promptable). See Figure 8.

• `speed_category`: The speed labeling function $\lambda_{\text{speed}}$ bins the scalar speed. For each timestep $t'$ we compute the speed $v_{t'}$ and uniformly partition $[0.5, 3.0]$ (real units) into $K = 3$ bins. With window radius $w(\lambda_{\text{speed}})$, the label at $t$ is the majority speed bin over $\{v_{t'}\}_{t' \in \tau_{t-w(\lambda_{\text{speed}})+1:t+w(\lambda_{\text{speed}})}}$. Hence $\mathcal{L}(\lambda_{\text{speed}}) = [\![0, K-1]\!]$. In practice we take $w(\lambda_{\text{speed}}) = 1$ to mitigate unnecessary credit assignment issues. We plot in Figure 9 the corresponding visuals and histograms.

• curvature_noise: The curvature-noise labeling function $\lambda_{\text{noise}}$ computes a variability statistic on a centered window. With unwrapped heading $(\theta_t)_t$, we define $\Delta\theta_{t'} = \theta_{t'+1} - \theta_{t'}$ and $\Delta^2\theta_{t'} = \Delta\theta_{t'+1} - \Delta\theta_{t'}$. On an odd window of size $W(\lambda_{\text{noise}})$ (default size 51) centered at $t$, we take $\sigma_t$ the standard deviation of the window's $(\Delta^2\theta_{t'})_t$ and uniformly bin $\sigma_t$ into $K = 3$ categories over $[0.0, 0.8]$. Hence $\mathcal{L}(\lambda_{\text{noise}}) = [\![0, K-1]\!]$. We plot in Figure 10 its visuals and histograms.

**Remark.** For all labels that use windows, the implementation ensures an odd, centered window around $t$, where relevant, "straight"/"undetermined" classes are excluded from promptable labels but kept in $\mathcal{L}(\lambda)$ for completeness. Bin edges are uniform by default and configurable through the class constructors.

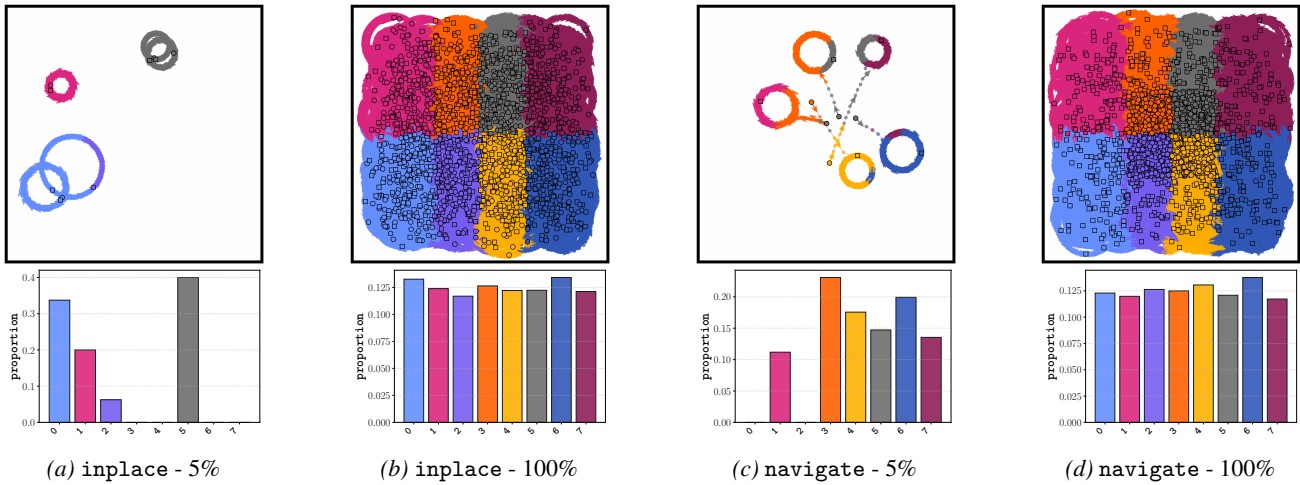

*Figure 5.* **Circle2d position label visualizations at different percentages.**

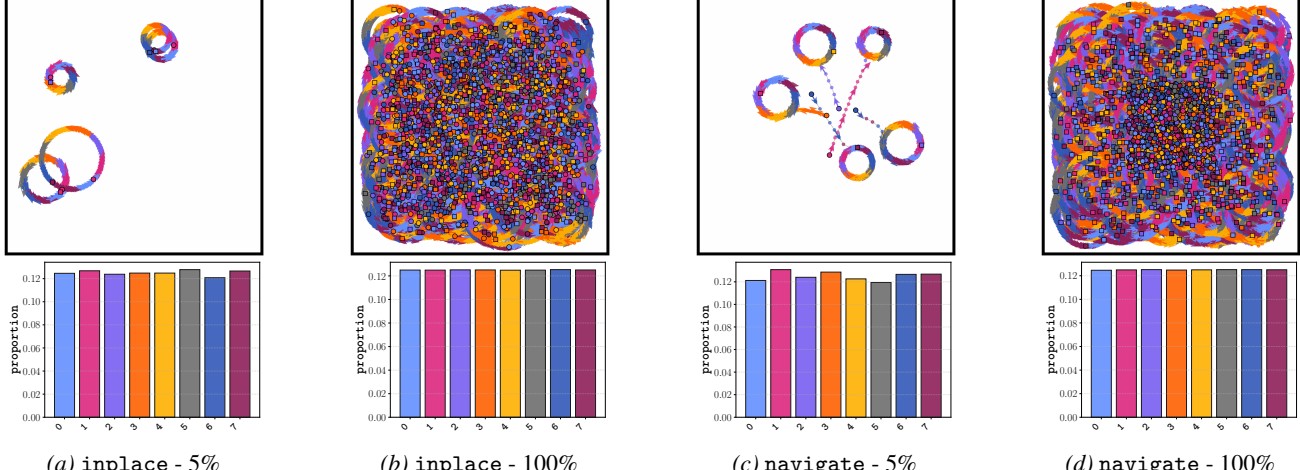

*Figure 6.* **Circle2d movement_direction label visualizations at different percentages.**

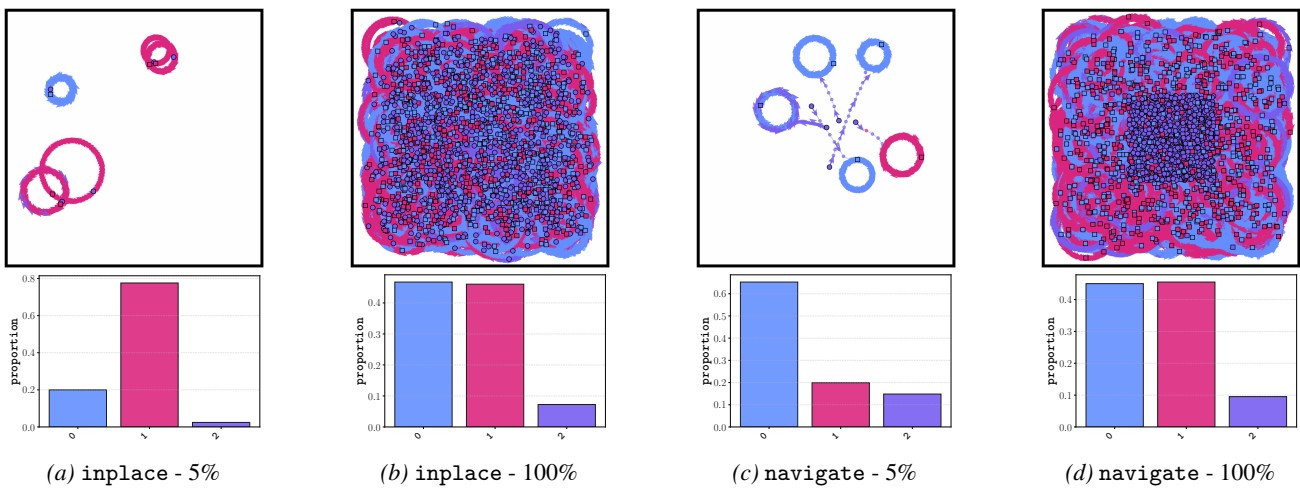

*(a)* `inplace` - 5%    *(b)* `inplace` - 100%    *(c)* `navigate` - 5%    *(d)* `navigate` - 100%

*Figure 7.* **Circle2d** `turn_direction` **label visualizations at different percentages.**

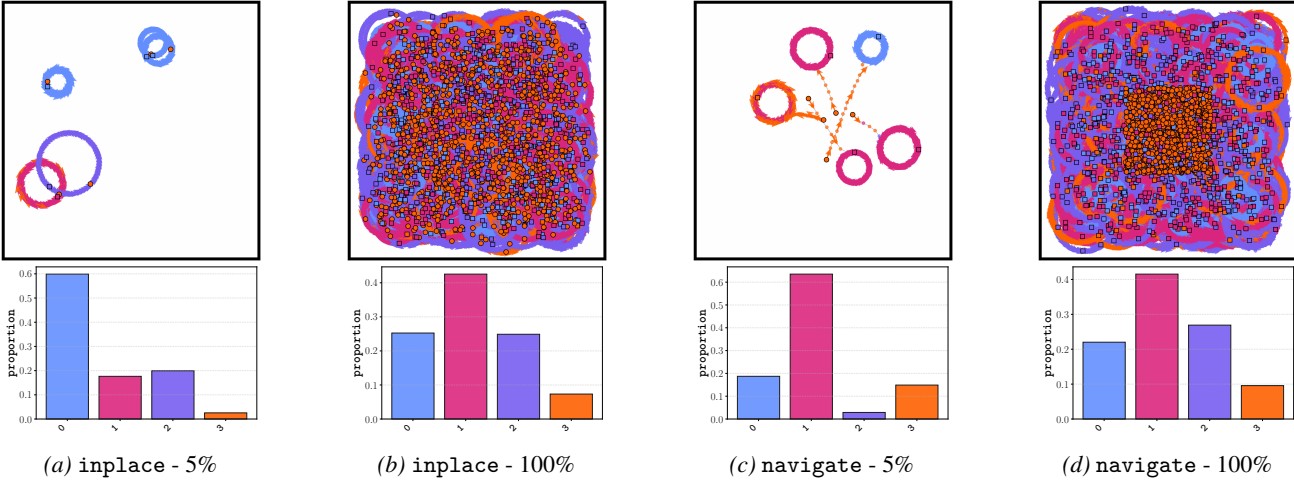

*(a)* `inplace` - 5%    *(b)* `inplace` - 100%    *(c)* `navigate` - 5%    *(d)* `navigate` - 100%

*Figure 8.* **Circle2d** `radius` **label visualizations at different percentages.**

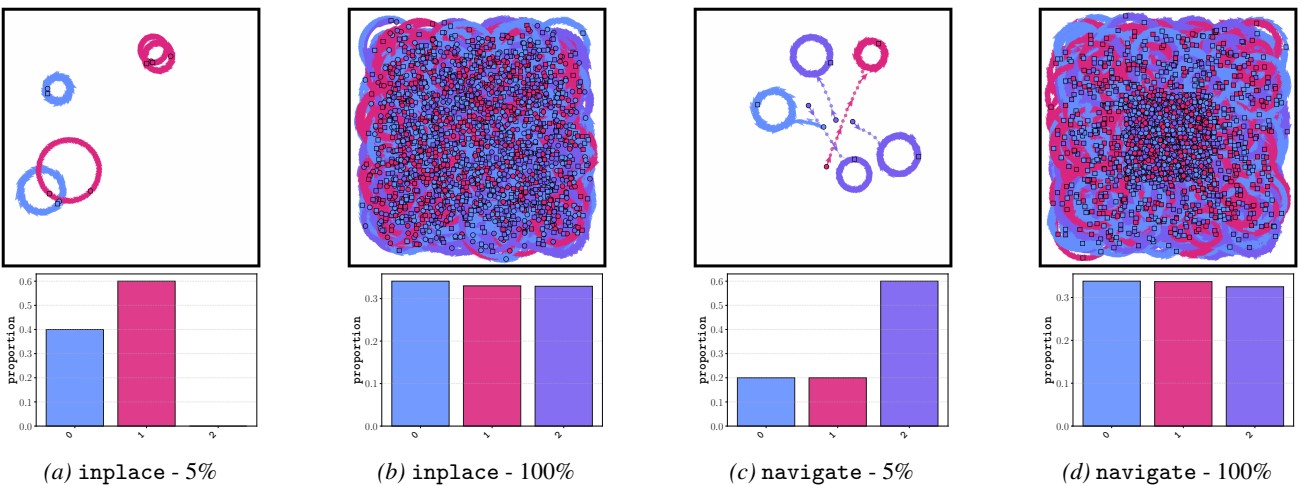

*(a)* `inplace` - 5%    *(b)* `inplace` - 100%    *(c)* `navigate` - 5%    *(d)* `navigate` - 100%

*Figure 9.* **Circle2d** `speed` **label visualizations at different percentages.**

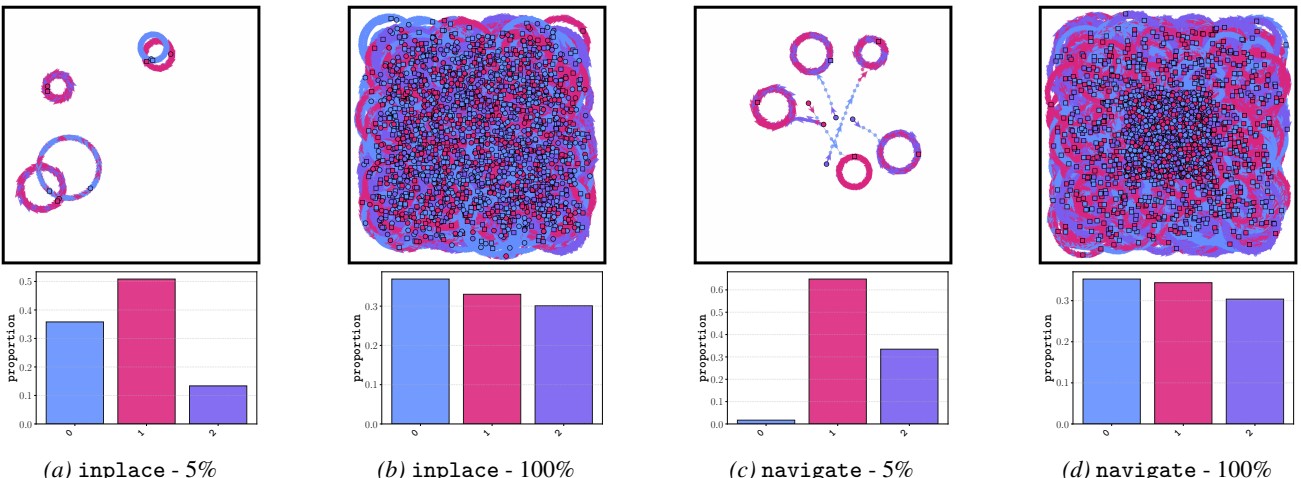

*(a)* `inplace` - 5%   *(b)* `inplace` - 100%   *(c)* `navigate` - 5%   *(d)* `navigate` - 100%

*Figure 10.* **Circle2d** `curvature_noise` **visualizations at different percentages.**

## A.2. HalfCheetah

**Environment** **HalfCheetah** (Todorov et al., 2012; Towers et al., 2024) is an environment consisting of controlling a 6-DoF 2-dimensional robot composed of 9 body parts and 8 joints connecting them. The environment has a time limit of 1000 timesteps. Details about this environment can be read in (Towers et al., 2024).

**Task** As implemented in (Towers et al., 2024), at each timestep $t$, the agent applies continuous control actions $a_t \in \mathbb{R}^d$ that drive the joints of the cheetah. The environment evaluates performance using a reward which encourages rapid forward progress while penalizing excessive control effort. Formally, the forward velocity of the torso is

$$v_t = \frac{x_{t+1} - x_t}{\Delta t},$$

where $x_t$ is the torso position along the horizontal axis and $\Delta t$ is the simulator timestep. The reward combines a positive term proportional to forward velocity with a quadratic control penalty:

$$r_t \;=\; w_f\, v_t \;-\; w_c \sum_{i=1}^{d} a_{t,i}^2,$$

where $w_f$ is the forward-reward weight and $w_c$ is the control-cost weight. Thus, the agent must learn to run efficiently: moving forward quickly while keeping joint torques as small as possible.

**Datasets** To generate the datasets, we train a diverse set of **HalfCheetah** policies through SAC (Haarnoja et al., 2018). We construct several *archetype* policies defined by Gaussian-shaped reward functions that bias behavior toward specific styles. The **Height** archetype rewards the torso maintaining a target vertical position $z_{\text{torso}}$ at specified values, thereby inducing qualitatively distinct gaits: *crawling* ($z \approx 0.5$ with $\sigma = 0.04$), *normal running* ($z \approx 0.6$ with $\sigma = 0.04$), or *upright running* ($z \approx 0.7$ with $\sigma = 0.04$). The **Speed** archetype rewards locomotion close to a desired forward velocity, producing policies that move at *slow pace* ($v \approx 1.5$), *medium pace* ($v \approx 5.0$), or *fast pace* ($v \approx 10.0$). Finally, the **Angle** archetype shapes behavior around the torso pitch angle, leading to policies that prefer *upright* ($\theta \approx -0.2$ with $\sigma = 0.05$), *flat* ($\theta \approx 0.0$ with $\sigma = 0.05$), or *crouched* ($\theta \approx 0.2$ with $\sigma = 0.05$) postures while still advancing forward. These archetypes yield a diverse collection of locomotion styles that serve as structured variations of the base **HalfCheetah** task. Then, we generate three datasets: `halfcheetah-fix-v0`, where the archetype policy is fixed during the trajectory, `halfcheetah-stitch-v0`, where the trajectories are cut into shorter segments from the `halfcheetah-fix-v0` dataset, and `halfcheetah-vary-v0`, where the policy archetype changes within the same trajectory. Each dataset contains $10^6 = 1000(\text{episodes}) * 1000(\text{timesteps})$ steps, with the `stitch` datasets containing more episodes as it cuts the `fix` dataset's episodes.

**Criteria and labels**  We present below the various labeling functions we designed for **HalfCheetah**. Each labeling function $\lambda$ maps raw environment signals to a discrete label sequence, optionally smoothed by a majority vote over a window $\tau_{t-w(\lambda)+1:t+w(\lambda)}$. In practice, we take $w(\lambda) = 1$ to mitigate unnecessary credit assignment issues.

- `speed`: The speed labeling function $\lambda_{\mathrm{speed}}$ discretizes the forward velocity magnitude $|v_t|$. We define a range $[v_{\min}, v_{\max}] = [0.1, 10.0]$ (real units) and split it uniformly into $K = 3$ bins, yielding the labels $\mathcal{L}(\lambda_{\mathrm{speed}}) = [\![0, 2]\!]$. We assign to step $t$ the majority index among the $(|v_{t'}|)_{t'}$'s bins of the window $\tau_{t-w(\lambda_{\mathrm{speed}})+1:t+w(\lambda_{\mathrm{speed}})}$. See Figure 11.

- `angle`: The angle labeling function $\lambda_{\mathrm{angle}}$ discretizes the torso pitch $\theta_t$. We define $[\theta_{\min}, \theta_{\max}] = [-0.3, 0.3]$ (radians) and split uniformly into $K = 3$ bins, yielding the label set $\mathcal{L}(\lambda_{\mathrm{angle}}) = [\![0, 2]\!]$. We assign to step $t$ the majority index among the $(\theta_{t'})_{t'}$'s bins of the window $\tau_{t-w(\lambda_{\mathrm{angle}})+1:t+w(\lambda_{\mathrm{angle}})}$. See Figure 12.

- `torso_height`: The torso-height labeling function $\lambda_{\mathrm{torso}}$ discretizes the vertical torso position $h_t$. We define $[h_{\min}, h_{\max}] = [0.4, 0.8]$ (real units) and split into $K = 3$ bins, giving $\mathcal{L}(\lambda_{\mathrm{torso}}) = [\![0, 2]\!]$. We assign to step $t$ the majority index among the $(h_{t'})_{t'}$'s bins of the window $\tau_{t-w(\lambda_{\mathrm{torso}})+1:t+w(\lambda_{\mathrm{torso}})}$. See Figure 13.

- `backfoot_height`: The back-foot labeling function $\lambda_{\mathrm{bf}}$ discretizes the vertical position of the back foot $h_t^{\mathrm{bf}}$. We define $[h_{\min}^{\mathrm{bf}}, h_{\max}^{\mathrm{bf}}] = [0.0, 0.3]$ and split into $K = 4$ bins, giving $\mathcal{L}(\lambda_{\mathrm{bf}}) = [\![0, 3]\!]$. We assign to step $t$ the majority index among the $(h_{t'}^{\mathrm{bf}})_{t'}$'s bins of the window $\tau_{t-w(\lambda_{\mathrm{bf}})+1:t+w(\lambda_{\mathrm{bf}})}$. See Figure 14.

- `frontfoot_height`: The front-foot labeling function $\lambda_{\mathrm{ff}}$ discretizes the vertical position of the front foot $h_t^{\mathrm{ff}}$ in the same manner as the back-foot: $[0.0, 0.3]$ split into $K = 4$ bins, yielding $\mathcal{L}(\lambda_{\mathrm{ff}}) = [\![0, 3]\!]$. We assign to step $t$ the majority index among the $(h_{t'}^{\mathrm{ff}})_{t'}$'s bins of the window $\tau_{t-w(\lambda_{\mathrm{ff}})+1:t+w(\lambda_{\mathrm{ff}})}$. See Figure 15.

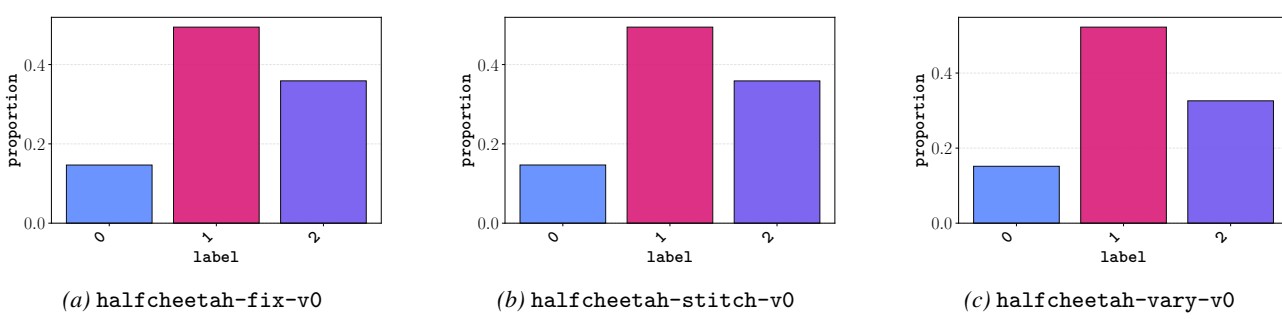

*(a)* `halfcheetah-fix-v0`    *(b)* `halfcheetah-stitch-v0`    *(c)* `halfcheetah-vary-v0`

*Figure 11.* **HalfCheetah** `speed` **label histograms.**

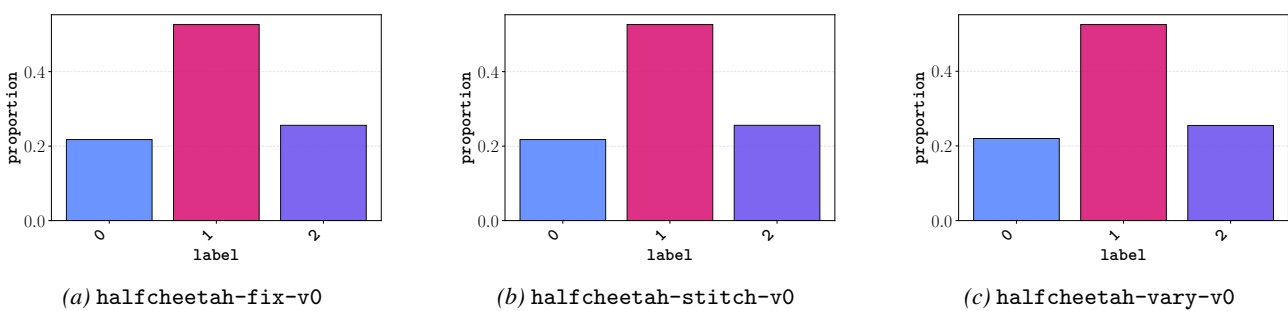

*(a)* `halfcheetah-fix-v0`    *(b)* `halfcheetah-stitch-v0`    *(c)* `halfcheetah-vary-v0`

*Figure 12.* **HalfCheetah** `angle` **label histograms.**

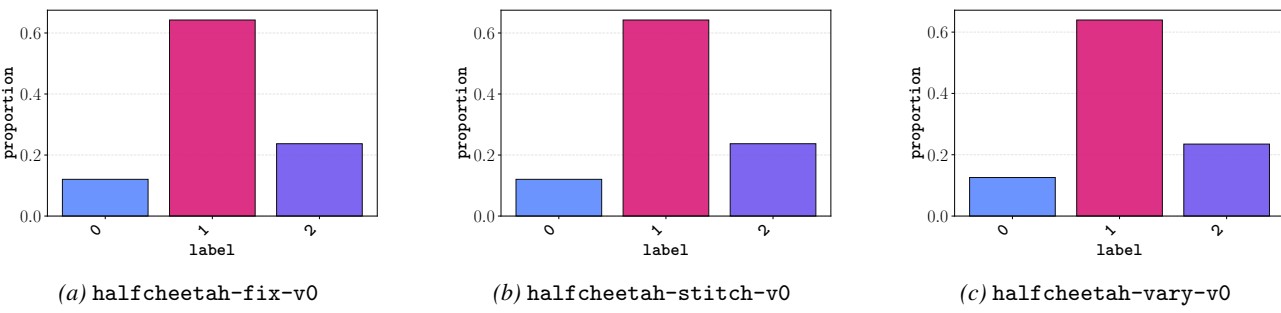

*(a)* `halfcheetah-fix-v0`     *(b)* `halfcheetah-stitch-v0`     *(c)* `halfcheetah-vary-v0`

*Figure 13.* **HalfCheetah** `torso_height` **label histograms.**

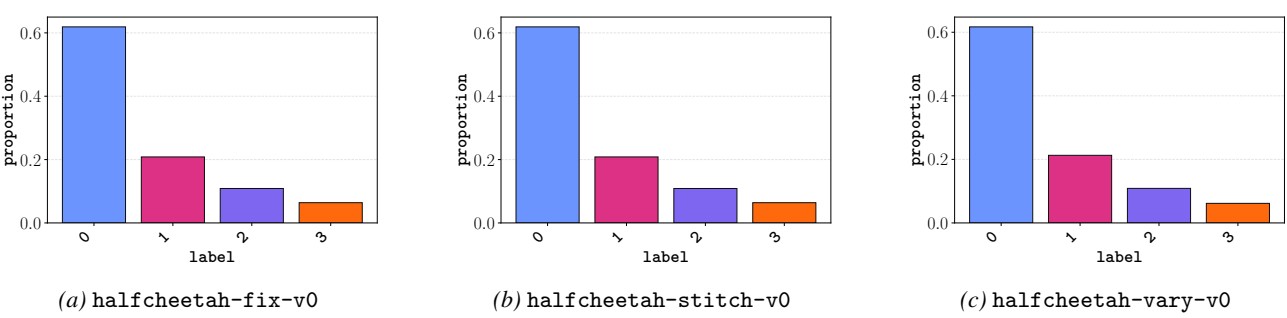

*(a)* `halfcheetah-fix-v0`     *(b)* `halfcheetah-stitch-v0`     *(c)* `halfcheetah-vary-v0`

*Figure 14.* **HalfCheetah** `backfoot_height` **label histograms.**

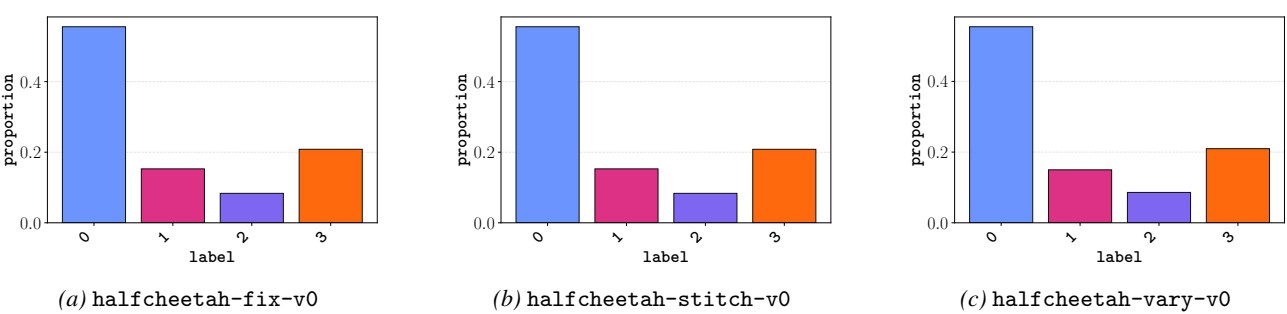

*(a)* `halfcheetah-fix-v0`     *(b)* `halfcheetah-stitch-v0`     *(c)* `halfcheetah-vary-v0`

*Figure 15.* **HalfCheetah** `frontfoot_height` **label histograms.**

### A.3. HumEnv

**Environment**   The **HumEnv** environment (Tirinzoni et al., 2025) is built on the SMPL skeleton (Loper et al., 2015), which consists of 24 rigid bodies, among which 23 are actuated. This SMPL skeleton is widely used in character animation and is well suited for expressing natural human-like stylized behaviors. **HumEnv**'s observations consist of the concatenation of the body poses (70 D), body rotations (144 D) and angular velocities (144D) resulting in a 358-dimensional vector. It moves the body using a proportional derivative controller resulting in a 69-dimensional action space. Consequently, this task has a higher dimensionality of (358, 69) compared to **HalfCheetah**'s (17, 6) dimensionality. We consider two types of **HumEnv** environments, **HumEnv-Simple**, which initializes the humanoid in a standing position, and **HumEnv-Complex**, which initializes the humanoid in a lying down position.

**Task**   At each timestep $t$, the agent applies continuous control actions $a_t \in \mathbb{R}^d$. The environments evaluate performance using a reward that encourages high-speed movement in the horizontal plane, modulated by a control efficiency term. Formally, let $v_{t,xy}$ denote the velocity vector of the center of mass projected onto the horizontal plane (ignoring vertical movement). The reward is defined as the norm of this velocity, scaled by a multiplicative control factor:

$$r_t = \alpha(a_t) \cdot \|v_{t,xy}\|_2,$$

where $\alpha(a_t) \in [0.8, 1.0]$ is a smoothness coefficient derived from a quadratic tolerance function on the control inputs $a_t$ provided in (Tirinzoni et al., 2025).

**Datasets**  We generated for each environment a stylized dataset using the Metamotivo-M1 model provided in (Tirinzoni et al., 2025), using various ways of moving at different heights and speeds. Since, the Metamotivo-M1 model was trained with a regularization toward the AMASS motion-capture dataset (Mahmood et al., 2019), it provides more natural and human-like stylized behaviors.

**Criteria and labels**  We present below the various labeling functions we designed for the **HumEnv** environments. Each labeling function $\lambda$ maps raw environment signals to a discrete label sequence, optionally smoothed by a majority vote over a window $\tau_{t-w(\lambda)+1:t+w(\lambda)}$. In practice, we take $w = 1$ to mitigate unnecessary credit assignment issues.

- `simple-head_height`: For **HumEnv-Simple** we focused our study on a single `head_height` criterion of two labels, namely `low` and `high`. The `simple-head-height` labeling function discretizes the vertical head position $h_t$ using a single threshold at 1.2. This results in $K = 2$ bins ($h_t < 1.2$ and $h_t \geq 1.2$), yielding the label set $\mathcal{L}(\lambda_{\text{simple\_head}}) = [\![0, 1]\!]$. See Figure 16a.

- `complex-speed`: For the **HumEnv-Complex**, we added a new `speed` criterion. The speed labeling function $\lambda_{\text{speed}}$ discretizes the center-of-mass velocity magnitude $|v_t|$. Based on the agent's movement capabilities, we define three distinct regimes: immobile ($|v_t| < 0.2$), slow ($0.2 \leq |v_t| \leq 3.0$), and fast ($|v_t| > 3.0$). This yields $K = 3$ bins with labels $\mathcal{L}(\lambda_{\text{speed}}) = [\![0, 2]\!]$. See Figure 16b.

- `complex-head_height`: For the **HumEnv-Complex**, we also extended the `complex-head_height` criterion by adding a new label for a total of 3 labels. The `head-height` labeling function $\lambda_{\text{complex\_head}}$ discretizes the vertical position of the agent's head $h_t$. We define thresholds at 0.4 and 1.2 to capture different postures: lying down, crouching and standing. The space is split into $K = 3$ bins: $h_t < 0.4$, $0.4 \leq h_t \leq 1.2$, and $h_t > 1.2$, yielding $\mathcal{L}(\lambda_{\text{complex\_head}}) = [\![0, 2]\!]$. See Figure 16c.

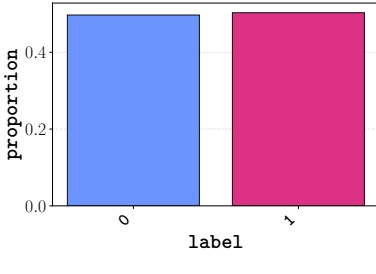
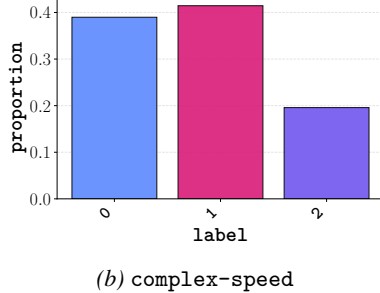
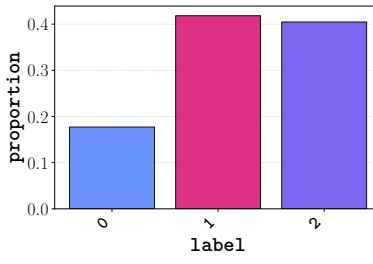

*(a)* `simple-head_height`  *(b)* `complex-speed`  *(c)* `complex-head_height`

*Figure 16.* **HumEnv label histograms.**

# B. Architectures and Hyperparameters

**Optimization:** For all baselines, when necessary, labels are encoded as latent variables of dimension 16 via an embedding matrix. We optimize all networks using the Adam optimizer with a learning rate of $3 \cdot 10^{-4}$, employing cosine learning-rate decay for the policies, a batch size of 256, and $10^5$ gradient steps for the $\chi$ estimators and $10^6$ for the other networks. Value functions $V$ additionally use layer normalization. Unless otherwise specified, we use the IQL hyperparameters $\beta = 3$, $\kappa = 0.7$, and $\gamma = 0.99$, and perform Polyak averaging on the $Q$-networks with coefficient $\upsilon_{\text{Polyak}} = 0.005$.

**Architectures:** For **Circle2d** and **HalfCheetah**, the policies $\pi$, value networks $V, Q$, and estimators $\chi$ are MLPs with hidden size $[256, 256]$ and ReLU activations. For **HumEnv**, the policies are MLPs with hidden size $[1024, 1024, 1024]$ and ReLU activations. When needed, we clip the advantages under 100 for the AWR loss computations.

**Implementations:** Our implementations are written in JAX (Bradbury et al., 2018), and take inspiration from (Nishimori, 2024), allowing short training durations. In **Circle2D** and **HalfCheetah**, we get for BC ($\approx 2$min), CBC ($\approx 3$min), BCPMI ($\approx 4$min), SORL ($\approx 15$min), SCBC ($\approx 3$min) and SCIQL ($\approx 35$min) on a NVIDIA V100 GPU for training runs. In **HumEnv**, we get for BC ($\approx 5$min), CBC ($\approx 5$min), BCPMI ($\approx 6$min), SORL ($\approx 23$min), SCBC ($\approx 5$min) and SCIQL ($\approx 45$min) on a NVIDIA A100 GPU for training runs. Our code and datasets can be found on our project website: `https://mathieu-petitbois.github.io/projects/sciql/`.

## C. Baselines

In this section, we describe in more detail our baselines. For the following, we define several sampling distributions on the unlabeled dataset $\mathcal{D}$ and labeled dataset $\lambda(\mathcal{D})$:

- $p^{\mathcal{D}}(s, a)(= p^{\lambda(\mathcal{D})}(s, a))$: Uniform sampling on $\mathcal{D}$'s state-action pairs

- $p_c^{\lambda(\mathcal{D})}(z \mid s, a)(= p^{\lambda(\mathcal{D})}(z \mid s, a))$: Dirac distribution of the style label of the **current** $(s, a)$ in its trajectory in $\lambda(\mathcal{D})$

- $p_f^{\lambda(\mathcal{D})}(z \mid s, a)$: Uniform distribution on the styles of the **future** $(s', a')$ pairs in $\lambda(\mathcal{D})$ starting from $(s, a)$

- $p_r^{\lambda(\mathcal{D})}(z)$: Uniform distribution of the style labels over the entire dataset $\lambda(\mathcal{D})$, which amounts to **random** selection

- $p_m^{\lambda(\mathcal{D})}(z \mid s, a)$: **Mixture** of $p_c^{\lambda(\mathcal{D})}(z \mid s, a)$, $p_f^{\lambda(\mathcal{D})}(z \mid s, a)$ and $p_r^{\lambda(\mathcal{D})}(z)$

**Behavior Cloning (BC).** BC (Pomerleau, 1991) is the simplest of our baselines and learns by maximizing the likelihood of actions given states through supervised learning on $\mathcal{D}$:

$$J_{\mathrm{BC}}(\pi) = \mathbb{E}_{(s,a) \sim p^{\mathcal{D}}(s,a)}[\log \pi(a \mid s)]. \tag{17}$$

We use this baseline as a reference for style alignment performance without conditioning.

**Conditioned Behavior Cloning (CBC).** CBC is the simplest style-conditioned method of our baselines and consists in concatenating to BC's states their associated label within $\lambda(\mathcal{D})$:

$$J_{\mathrm{CBC}}(\pi) = \mathbb{E}_{(s,a) \sim p^{\mathcal{D}}(s,a), z \sim p_c^{\mathcal{D}}(z \mid s,a)}[\log \pi(a \mid s, z)] \tag{18}$$

This baseline serves as a reference to test the various benefits of subsequent methods to better perform style alignment optimization.

**Behavior Cloning with Pointwise Mutual Information weighting (BCPMI).** BCPMI (Yang et al., 2025) seeks to address credit assignment issues between state–action pairs and style labels by relying on their mutual information estimates. For this, BCPMI uses Mutual Information Neural Estimation (MINE). In the information-theoretic setting, let $S$, $A$, and $Z$ be random variables corresponding to states, actions, and styles, respectively. The mutual information between state–action pairs $(S, A)$ and styles $Z$ can be written as the Kullback–Leibler (KL) divergence between the joint distribution $P_{S,A,Z}$ and the product of their marginals $P_{S,A} \otimes P_Z$:

$$I(S, A; Z) = D_{KL}(P_{S,A,Z} \parallel P_{S,A} \otimes P_Z). \tag{19}$$

As directly estimating this mutual information is difficult, MINE relies on the Donsker-Varadhan lower bound:

$$I(S, A; Z) \geq \sup_{T \in \mathcal{F}} \mathbb{E}_{(s,a,z) \sim P_{S,A,Z}}[T(s, a, z)] - \log\left(\mathbb{E}_{(s,a,z) \sim P_{S,A} \otimes P_Z}[e^{T(s,a,z)}]\right), \tag{20}$$

where $\mathcal{F}$ denotes a class of functions $T : \mathcal{S} \times \mathcal{A} \times \mathcal{Z} \to \mathbb{R}$. According to (Donsker & Varadhan, 1975), optimizing this bound yields

$$T^*(s, a, z) = \log \frac{p(s, a, z)}{p(s, a)p(z)} = \log \frac{p(z \mid s, a)}{p(z)}. \tag{21}$$

This is particularly true for $T_{\pi_{\mathcal{D}}}^{\lambda,*}$ associated with the dataset's distribution $p_{\pi_{\mathcal{D}}}^{\lambda}$. BCPMI trains a neural network to approximate $T_{\pi_{\mathcal{D}}}^{\lambda,*}$ and uses it to weight CBC's learning objective, increasing the impact of transitions with high style relevance while reducing that of less relevant ones:

$$J_{\mathrm{MINE}}(T_{\pi_{\mathcal{D}}}^{\lambda}) = \mathbb{E}_{(s,a) \sim p^{\mathcal{D}}(s,a),\, z \sim p_c^{\lambda(\mathcal{D})}(z \mid s,a)}[T_{\pi_{\mathcal{D}}}^{\lambda}(s, a, z)] - \log\left(\mathbb{E}_{(s,a) \sim p^{\mathcal{D}}(s,a),\, z \sim p_r^{\lambda(\mathcal{D})}(z)}[e^{T_{\pi_{\mathcal{D}}}^{\lambda}(s,a,z)}]\right), \tag{22}$$

$$J_{\mathrm{BC-PMI}}(\pi) = \mathbb{E}_{(s,a) \sim p^{\mathcal{D}}(s,a),\, z \sim p_c^{\lambda(\mathcal{D})}(z \mid s,a)}[\exp(T_{\pi_{\mathcal{D}}}^{\lambda,*}(s, a, z)) \log \pi(a \mid s, z)]. \tag{23}$$

This baseline is notable as it constitutes a first step toward addressing the credit assignment challenges in style-conditioned policy learning. However, as it strictly focuses on imitation learning rather than task performance, it does not support style mixing and is therefore not designed to address distribution shifts at inference time, unlike our method.

**Stylized Offline Reinforcement Learning (SORL):** SORL (Mao et al., 2024) is an important baseline to consider since it both addresses the optimization of policy diversity and task performance. Initially designed within an unsupervised learning setting, SORL is a two step algorithm which aims to learn a diverse set of high-performing policies from $\mathcal{D}$. First, SORL uses the Expectation-Maximisation (EM) algorithm to first learn a finite set of diverse policies $\{\mu^{(i)}\}_{i=1}^{N}$ to capture the heterogeneity of $\mathcal{D}$. The E step aims to fit an estimate $\hat{p}(z = i \mid \tau)$ to the posteriors $p(z = i \mid \tau)$, associating each trajectory to a given style among $N$ styles. The M step aims to train the stylized policies $\{\mu^{(i)}\}_{i=1}^{N}$ according to their associated style through $\hat{p}(z = i \mid \tau)$:

$$\underline{\text{E step:}} \ \forall i \in \{1, ..., N\}, \hat{p}(z = i \mid \tau) \approx \frac{1}{Z} \sum_{(s,a) \in \tau} \mu^{(i)}(a \mid s) \text{ where } Z \text{ is a normalizing factor} \tag{24}$$

$$\underline{\text{M step:}} \ J_{\text{SORL - M step}}(\{\mu^{(i)}\}_{i=1}^{N}) = \frac{1}{|\mathcal{D}|} \sum_{\tau \in \mathcal{D}} \sum_{i=1}^{N} \hat{p}(z = i \mid \tau) \sum_{(s,a) \in \tau} \log \mu^{(i)}(a \mid s) \tag{25}$$

Then, to perform task-performance optimization while preserving a certain amount of diversity, SORL proposes to train from $\{\mu^{(i)}\}_{i=1}^{N}$ a set of policies $\{\pi^{(i)}\}_{i=1}^{N}$ by solving the following constrained RL problem:

$$\forall i \in \{1, ..., N\}, \quad \pi^{(i)} = \arg\max_{\pi} J(\pi) \tag{26}$$

$$\text{s.t.} \quad \mathbb{E}_{s \sim \rho_{\mu^{(i)}}(s)} D_{KL}\big(\pi^{(i)}(\cdot \mid s) \parallel \mu^{(i)}(\cdot \mid s)\big) \leq \epsilon, \quad \int_{a} \pi^{(i)}(a \mid s) \, da = 1, \forall s, \tag{27}$$

with $\rho_{\pi}$ the unnormalized discounted state distribution of a policy $\pi$. By using its associated Lagrangian optimization problem, (Mao et al., 2024) show that this problem can be cast into a Stylized Advantage Weighted Regression (SAWR) objective:

$$\forall i \in \{1, ..., N\}, J_{\text{SORL - SAWR}}(\pi^{(i)}) = \mathbb{E}_{\tau \sim \mathcal{D}} \hat{p}(z = i \mid \tau) \sum_{(s,a) \in \tau} \log \pi^{(i)}(a \mid s) \exp\left(\frac{1}{\alpha} A^{\mu}(s, a)\right). \tag{28}$$

with $A^{\mu}(s, a)$ the advantage relating to the dataset's distribution. In our supervised setting, given a criterion $\lambda$, the first step translates into the learning of a style-conditioned policy $\mu^{*} : \mathcal{S} \to \Delta(\mathcal{A}) \in \arg\max_{\mu} S^{p}(\mu, \lambda, z), \forall z \in \mathcal{L}(\lambda)$ by optimizing the style alignment objective while the second step translates into optimizing $\mu^{*}$'s performance by learning the solution $\pi^{*}$ of the following constrained problem:

$$\forall z \in \mathcal{L}(\lambda), \pi^{*}(\cdot \mid \cdot, z) = \underset{\pi(\cdot \mid \cdot, z)}{\text{argmax}} J(\pi(\cdot \mid \cdot, z)) \tag{29}$$

$$\text{s.t. } \mathbb{E}_{s \sim \rho_{\mu^{*}(\cdot \mid \cdot, z)}(s)} D_{KL}(\pi(\cdot \mid s, z) \| \mu^{*}(\cdot \mid s, z)) \leq \varepsilon, \int_{a} \pi(a \mid s, z) da = 1, \forall s \tag{30}$$

Let $z \in \mathcal{L}(\lambda) = \{0, ..., |\lambda| - 1\}$ be a discrete style label. Following a similar path as (Peng et al., 2019) and (Mao et al., 2024), we can state that maximizing $J(\pi(\cdot \mid \cdot, z))$ is similar to maximizing the expected improvement $\eta(\pi(\cdot \mid \cdot, z)) = J(\pi(\cdot \mid \cdot, z)) - J(\mu^{*}(\cdot \mid \cdot, z))$, which can be expressed as (Schulman et al., 2015) show as:

$$\eta(\pi(\cdot \mid \cdot, z)) = \mathbb{E}_{s \sim \rho_{\pi(\cdot \mid \cdot, z)}(s)} \mathbb{E}_{a \sim \pi(\cdot \mid s, z)}[A^{\mu^{*}(\cdot \mid \cdot, z)}(s, a)] \tag{31}$$

Like (Peng et al., 2019) showed, we can substitute $\rho_{\pi(\cdot \mid \cdot, z)}$ for $\rho_{\mu^{*}(\cdot \mid \cdot, z)}$ to simplify this optimization problem as the resulting error has been shown to be bounded by $D_{KL}(\pi(\cdot \mid \cdot, z) \| \mu^{*}(\cdot \mid \cdot, z))$ (Schulman et al., 2015). Furthermore, (Peng et al., 2019) and (Mao et al., 2024) approximate $A^{\mu^{*}(\cdot \mid \cdot, z)}(s, a)$ by the advantage $A^{\mu}(s, a)$ where $\mu$ represents the policy distribution of the dataset. In our setting, we will use the advantage $A^{r}(s, a)$ estimated through IQL to be coherent with SCIQL. Consequently, SORL's stylized advantage weighted regression becomes in our context:

$$\pi^{*}(\cdot \mid \cdot, z) = \underset{\pi(\cdot \mid \cdot, z)}{\text{argmax}} \mathbb{E}_{s \sim \rho_{\mu^{*}(\cdot \mid \cdot, z)}(s)} \mathbb{E}_{a \sim \pi(\cdot \mid s, z)}[A^{r}(s, a)] \tag{32}$$

$$\text{s.t. } \mathbb{E}_{s \sim \rho_{\mu^{*}(\cdot \mid \cdot, z)}(s)} D_{KL}(\pi(\cdot \mid s, z) \| \mu^{*}(\cdot \mid s, z)) \leq \varepsilon, \int_{a} \pi(a \mid s, z) da = 1, \forall s \tag{33}$$

As Peng et al. (2019) and Mao et al. (2024), we compute the corresponding Lagrangian of this optimization problem:

$$L(\pi(\cdot \mid \cdot, z), \alpha^{\mu^*}, \boldsymbol{\alpha}^\pi) = \mathbb{E}_{s \sim \rho_{\mu^*(\cdot \mid \cdot, z)}}\Big[\mathbb{E}_{a \sim \pi(\cdot \mid s, z)} A^r(s, a) \tag{34}$$

$$+ \alpha^{\mu^*}\big(\varepsilon - D_{KL}(\pi(\cdot \mid s, z) \,\|\, \mu^*(\cdot \mid s, z))\big)\Big] \tag{35}$$

$$+ \int_s \boldsymbol{\alpha}_s^\pi\Big(1 - \int_a \pi(a \mid s, z)\,da\Big)ds \tag{36}$$

$$= \int_s \rho_{\mu^*(\cdot \mid \cdot, z)}(s)ds\Big[\int_a \pi(a \mid s, z)da A^r(s, a) \tag{37}$$

$$+ \alpha^{\mu^*}\Big(\varepsilon - \int_a \pi(a \mid s, z) \log \frac{\pi(a \mid s, z)}{\mu^*(a \mid s, z)}da\Big] \tag{38}$$

$$+ \int_s \boldsymbol{\alpha}_s^\pi\Big(1 - \int_a \pi(a \mid s, z)\,da\Big)ds \tag{39}$$

with $\alpha^{\mu^*} \geq 0$ and $\boldsymbol{\alpha}^\pi = \{\boldsymbol{\alpha}_s^\pi \in \mathbb{R}, s \in \mathcal{S}\}$ the Lagrange multipliers. We differentiate $L(\pi(\cdot \mid \cdot, z), \alpha^{\mu^*}, \boldsymbol{\alpha}^\pi)$ as:

$$\frac{\partial L}{\partial \pi(a \mid s, z)} = \rho_{\mu^*(\cdot \mid s, z)}(s)\Big[A^r(s, a) - \alpha^{\mu^*} \log \pi(a \mid s, z) + \alpha^{\mu^*} \log \mu^*(a \mid s, z) - \alpha^{\mu^*}\Big] - \boldsymbol{\alpha}_s^\pi \tag{40}$$

Setting this derivative to zero yields the following closed-form solution:

$$\pi^*(a \mid s, z) = \frac{1}{Z(s, z)}\mu^*(a \mid s, z)\exp\Big(\frac{1}{\alpha^{\mu^*}}A^r(s, a)\Big), \tag{41}$$

where $Z(s, z)$ is the normalization term defined as:

$$Z(s, z) = \exp\Big(\frac{1}{\rho_{\mu^*(\cdot \mid \cdot, z)}(s)}\frac{\boldsymbol{\alpha}_s^\pi}{\alpha^{\mu^*}} + 1\Big). \tag{42}$$

Finally, as Peng et al. (2019) and Mao et al. (2024), we estimate $\pi^*(\cdot \mid \cdot, z)$ with a neural network policy $\pi_\psi(\cdot \mid \cdot, z)$ by solving:

$$\arg\min_\psi \mathbb{E}_{s \sim p^{\lambda(\mathcal{D})}(s \mid z)}\Big[D_{KL}\big(\pi^*(\cdot \mid s, z) \,\|\, \pi_\psi(\cdot \mid s, z)\big)\Big] \tag{43}$$

$$= \arg\min_\psi \mathbb{E}_{s \sim p^{\lambda(\mathcal{D})}(s \mid z)}\Big[\int_a \big(\pi^*(a \mid s, z)\log\pi^*(a \mid s, z) - \pi^*(a \mid s, z)\log\pi_\psi(a \mid s, z)\big)da\Big] \tag{44}$$

$$= \arg\min_\psi \ -\mathbb{E}_{s \sim p^{\lambda(\mathcal{D})}(s \mid z)}\Big[\int_a \pi^*(a \mid s, z)\log\pi_\psi(a \mid s, z)\,da\Big] \tag{45}$$

$$= \arg\min_\psi \ -\mathbb{E}_{s \sim p^{\lambda(\mathcal{D})}(s \mid z)}\Big[\int_a \frac{1}{Z(s, z)}\mu^*(a \mid s, z)\exp\big(\tfrac{1}{\alpha^{\mu^*}}A^r(s, a)\big)\log\pi_\psi(a \mid s, z)\,da\Big] \tag{46}$$

$$= \arg\min_\psi \ -\mathbb{E}_{(s,a) \sim p^{\lambda(\mathcal{D})}(s,a \mid z)}\Big[\frac{1}{Z(s, z)}\exp\big(\tfrac{1}{\alpha^{\mu^*}}A^r(s, a)\big)\log\pi_\psi(a \mid s, z)\Big] \tag{47}$$

$$= \arg\min_\psi \ -\mathbb{E}_{(s,a) \sim p^{\lambda(\mathcal{D})}(s,a)}\Big[p(z \mid s, a)\frac{1}{Z(s, z)}\exp\big(\tfrac{1}{\alpha^{\mu^*}}A^r(s, a)\big)\log\pi_\psi(a \mid s, z)\Big] \tag{48}$$

By neglecting the absorbing constant as (Peng et al., 2019; Mao et al., 2024), we can finally express the SORL objective in our supervised version:

$$\arg\min_\psi \ -\mathbb{E}_{(s,a) \sim p^{\lambda(\mathcal{D})}(s,a)}\Big[p(z \mid s, a)\,\exp\big(\tfrac{1}{\alpha^{\mu^*}}A^r(s, a)\big)\log\pi_\psi(a \mid s, z)\Big] \tag{49}$$

As we want to optimize this objective for all of the $z \in \mathcal{L}(\lambda) = \{0, ..., |\lambda| - 1\}$, we write below the general objective:

$$\arg\min_\psi \ -\mathbb{E}_{(s,a) \sim p^{\lambda(\mathcal{D})}(s,a)}\Big[\frac{1}{|\lambda|}\sum_{z=0}^{|\lambda|-1} p(z \mid s, a)\,\exp\big(\tfrac{1}{\alpha^{\mu^*}}A^r(s, a)\big)\log\pi_\psi(a \mid s, z)\Big] \tag{50}$$

As in SCIQL, we can employ several strategies to estimate $p(z \mid s, a)$ through an estimator $\chi(s, a, z)$ which we all detail in Appendix E.1. Additionally, the advantage functions can be learned offline through IQL as in SCIQL. Hence, we can obtain our adapted SORL objectives by taking $\beta = 1/\alpha^{\mu^*}$:

$$\mathcal{L}_{\text{SORL}}(V_r) = \mathbb{E}_{(s,a) \sim p^{\mathcal{D}}(s,a)}[\ell_\kappa^2(\bar{Q}_r(s,a) - V_r(s))] \tag{51}$$

$$\mathcal{L}_{\text{SORL}}(Q_r) = \mathbb{E}_{(s,a,s') \sim p^{\mathcal{D}}(s,a,s')}[(r(s,a) + \gamma V_r(s') - Q_r(s,a))^2] \tag{52}$$

$$J_{\text{SORL}}(\pi) = \mathbb{E}_{(s,a) \sim p^{\mathcal{D}}(s,a)} \frac{1}{|\lambda|} \sum_{z=0}^{|\lambda|-1} \chi(s,a,z) e^{\beta A^r(s,a)} \log \pi(a \mid s, z) \tag{53}$$

**Style-Conditioned Behavior Cloning (SCBC):** SCBC corresponds to a simpler behavior cloning version of SCIQL whose objective can be written as:

$$J_{\text{SCBC}}(\pi) = \mathbb{E}_{(s,a) \sim p^{\mathcal{D}}(s,a), z \sim p_{\text{f}}^{\mathcal{D}}(z|s,a)}[\log \pi(a \mid s, z)] \tag{54}$$

This baseline is interesting as it shows both how style mixing with hindsight relabeling can be beneficial to style alignment while highlighting the impact of value learning when compared to SCIQL. For instance, value learning allows for relabeling outside of $p_{\text{f}}^{\lambda(\mathcal{D})}$ on top of optimizing the policy.

# D. Additional tables

*Table 2.* **Experiment complexity:** In this table, we detail the experimental scale of our experiments, highlighting the need to perform averaging over seeds, criteria, labels and eval episodes in the results tables.

| Environment | Criterion | $n_{\text{labels}}$ | $n_{\text{datasets}}$ | $n_{\text{seeds}}$ | Total trainings | $n_{\text{eval\_episodes}}$ | Total eval episodes |
|---|---|---|---|---|---|---|---|
| circle2d | position | 8 | 2 | 5 | 80 | 10 | 800 |
| | movement_direction | 8 | 2 | 5 | 80 | 10 | 800 |
| | turn_direction | 2 | 2 | 5 | 20 | 10 | 200 |
| | radius | 3 | 2 | 5 | 30 | 10 | 300 |
| | speed | 3 | 2 | 5 | 30 | 10 | 300 |
| | curvature_noise | 3 | 2 | 5 | 30 | 10 | 300 |
| halfcheetah | speed | 3 | 3 | 5 | 45 | 10 | 450 |
| | angle | 3 | 3 | 5 | 45 | 10 | 450 |
| | torso_height | 3 | 3 | 5 | 45 | 10 | 450 |
| | backfoot_height | 4 | 3 | 5 | 60 | 10 | 600 |
| | frontfoot_height | 4 | 3 | 5 | 60 | 10 | 600 |
| humenv-simple | simple-head_height | 2 | 1 | 5 | 10 | 10 | 100 |
| humenv-complex | complex-speed | 3 | 1 | 5 | 15 | 10 | 150 |
| | complex-head_height | 3 | 1 | 5 | 15 | 10 | 150 |
| all | 14 criteria | 52 | _ | _ | 565 | _ | 5650 |

In this section, we display the full results for both style alignment and style-conditioned task-performance optimization. These tables are computed for each environment and criterion $\lambda$ by averaging performance across 5 seeds and all labels in $\mathcal{L}(\lambda)$. Table 2 reports the evaluation complexity statistics of our experiments, which, for each algorithm variant, requires 565 training runs and 5650 evaluation episodes. Normalized per seed, this corresponds to $565/5 = 113$ runs per algorithm, which justifies our use of averages in Table 1, Table 3, and Table 4. In the following, we write additional remarks about the full results tables.

**Style alignment (Table 3):** In Table 3, SCIQL achieves better style alignment on most criteria, while being slightly lower on the turn_direction, radius, and speed criteria of **Circle2d**. This can be explained by the fact that these criteria do not require relabeling, and we show in Appendix E.2 that optimal performance can be recovered by changing the sampling distribution from $p_r^{\lambda(\mathcal{D})}$ that we globally use to $p_c^{\lambda(\mathcal{D})}$ for those particular criteria. Additionally, methods that do not perform style relabeling perform worse in inplace than in navigate for style criteria corresponding to specific subsets of the state space, such as position, highlighting the importance of style relabeling for alignment. For **HalfCheetah**, SCIQL largely dominates all baselines demonstrating SCIQL's robustness to noisier trajectories. Namely, in the halfcheetah-vary-v0, SCIQL dominates even more the baselines. In particular, SCBC sees an important decrease in its style alignment. This can be explained by the nature of the relabeling used in SCBC, detailed Appendix C. For a given observed state-action pair in the dataset $(s, a)$, SCBC samples a future style label $z_f$ from the future of the trajectory and considers $(s, a, z_f)$ as expert behavior. Indeed, for SCBC, every action is expert to reach the styles in the future of its trajectory. However, when style variations occur within the trajectory, for instance when alternating low and high speeds $(z_{\text{slow}}, ..., z_{\text{fast}}, ..., z_{\text{slow}}, ...)$, an action contributing to high speed $(s, a, z_t)$ with $z_f = z_{\text{fast}}$ could be relabeled as $(s, a, z_{\text{slow}})$, provoking the learning of an action for high speeds while being conditioned on $z_{\text{slow}}$. SCIQL solves this problem by adding an advantage weighted regression mechanism to always strive to reach as fast as possible style alignment, consequently lowering the weights of wrong labels.

*Table 3.* **Style alignment results (full).**

| Dataset | BC | CBC | BCPMI | SORL ($\beta = 0$) | SCBC | SCIQL |
|---|---|---|---|---|---|---|
| `circle2d-inplace-v0 - position` | $12.5_{\pm6.9}$ | $15.0_{\pm10.3}$ | $16.3_{\pm13.5}$ | $14.9_{\pm11.6}$ | $65.9_{\pm11.5}$ | $98.0_{\pm0.3}$ |
| `circle2d-inplace-v0 - movement_direction` | $12.5_{\pm0.2}$ | $4.4_{\pm1.6}$ | $4.1_{\pm1.4}$ | $5.3_{\pm4.2}$ | $12.5_{\pm0.3}$ | $20.5_{\pm4.4}$ |
| `circle2d-inplace-v0 - turn_direction` | $50.0_{\pm25.1}$ | $100.0_{\pm0.0}$ | $100.0_{\pm0.1}$ | $100.0_{\pm0.1}$ | $100.0_{\pm0.0}$ | $82.6_{\pm26.3}$ |
| `circle2d-inplace-v0 - radius` | $33.3_{\pm1.2}$ | $99.1_{\pm2.0}$ | $99.7_{\pm0.6}$ | $99.8_{\pm0.4}$ | $100.0_{\pm0.0}$ | $96.1_{\pm5.3}$ |
| `circle2d-inplace-v0 - speed` | $33.3_{\pm4.2}$ | $99.9_{\pm0.1}$ | $99.9_{\pm0.0}$ | $99.9_{\pm0.0}$ | $99.9_{\pm0.0}$ | $91.6_{\pm13.3}$ |
| `circle2d-inplace-v0 - curvature_noise` | $33.3_{\pm0.0}$ | $33.3_{\pm0.0}$ | $33.3_{\pm0.1}$ | $33.3_{\pm0.0}$ | $33.3_{\pm0.0}$ | $59.1_{\pm6.1}$ |
| `circle2d-inplace-v0 - all` | $29.1_{\pm6.3}$ | $58.6_{\pm2.3}$ | $58.9_{\pm2.6}$ | $58.9_{\pm2.7}$ | $68.6_{\pm2.0}$ | $\mathbf{74.6}_{\pm9.3}$ |
| `circle2d-navigate-v0 - position` | $12.5_{\pm7.4}$ | $16.7_{\pm9.5}$ | $24.0_{\pm11.8}$ | $22.3_{\pm14.8}$ | $58.5_{\pm9.5}$ | $98.4_{\pm0.2}$ |
| `circle2d-navigate-v0 - movement_direction` | $12.5_{\pm0.2}$ | $5.7_{\pm4.9}$ | $3.2_{\pm0.2}$ | $4.9_{\pm3.7}$ | $12.5_{\pm0.2}$ | $27.0_{\pm5.7}$ |
| `circle2d-navigate-v0 - turn_direction` | $50.0_{\pm13.4}$ | $100.0_{\pm0.0}$ | $100.0_{\pm0.0}$ | $100.0_{\pm0.1}$ | $99.6_{\pm0.1}$ | $96.0_{\pm5.7}$ |
| `circle2d-navigate-v0 - radius` | $33.3_{\pm10.6}$ | $98.1_{\pm1.7}$ | $98.8_{\pm1.4}$ | $99.7_{\pm0.4}$ | $99.2_{\pm0.9}$ | $95.8_{\pm5.6}$ |
| `circle2d-navigate-v0 - speed` | $33.3_{\pm0.0}$ | $99.9_{\pm0.0}$ | $99.9_{\pm0.0}$ | $99.6_{\pm0.7}$ | $99.9_{\pm0.0}$ | $96.0_{\pm4.5}$ |
| `circle2d-navigate-v0 - curvature_noise` | $33.3_{\pm0.0}$ | $33.3_{\pm0.1}$ | $33.3_{\pm0.3}$ | $33.3_{\pm0.0}$ | $33.4_{\pm0.1}$ | $40.0_{\pm6.7}$ |
| `circle2d-navigate-v0 - all` | $29.1_{\pm5.3}$ | $58.9_{\pm2.7}$ | $59.9_{\pm2.3}$ | $60.0_{\pm3.3}$ | $67.2_{\pm1.8}$ | $\mathbf{75.5}_{\pm4.7}$ |
| `halfcheetah-fix-v0 - speed` | $33.3_{\pm11.2}$ | $73.9_{\pm11.8}$ | $77.6_{\pm9.0}$ | $73.0_{\pm20.3}$ | $95.9_{\pm1.2}$ | $96.0_{\pm1.6}$ |
| `halfcheetah-fix-v0 - angle` | $33.3_{\pm4.5}$ | $57.7_{\pm15.5}$ | $68.0_{\pm11.3}$ | $60.0_{\pm15.5}$ | $55.2_{\pm7.4}$ | $99.1_{\pm1.1}$ |
| `halfcheetah-fix-v0 - torso_height` | $33.3_{\pm6.0}$ | $70.9_{\pm11.1}$ | $82.2_{\pm10.0}$ | $73.2_{\pm8.9}$ | $79.3_{\pm8.3}$ | $96.8_{\pm3.5}$ |
| `halfcheetah-fix-v0 - backfoot_height` | $25.0_{\pm2.5}$ | $26.9_{\pm2.6}$ | $29.6_{\pm3.9}$ | $28.4_{\pm2.8}$ | $32.4_{\pm6.8}$ | $47.5_{\pm2.0}$ |
| `halfcheetah-fix-v0 - frontfoot_height` | $25.0_{\pm5.5}$ | $26.5_{\pm3.9}$ | $33.3_{\pm7.8}$ | $30.7_{\pm5.7}$ | $27.0_{\pm3.0}$ | $50.5_{\pm0.8}$ |
| `halfcheetah-fix-v0 - all` | $30.0_{\pm5.9}$ | $51.2_{\pm9.0}$ | $58.1_{\pm8.4}$ | $53.1_{\pm10.6}$ | $58.0_{\pm5.3}$ | $\mathbf{78.0}_{\pm1.8}$ |
| `halfcheetah-stitch-v0 - speed` | $33.3_{\pm8.7}$ | $79.9_{\pm8.0}$ | $70.1_{\pm17.7}$ | $57.1_{\pm23.2}$ | $92.0_{\pm3.3}$ | $96.3_{\pm0.5}$ |
| `halfcheetah-stitch-v0 - angle` | $33.3_{\pm8.0}$ | $50.4_{\pm14.2}$ | $72.1_{\pm18.9}$ | $55.0_{\pm20.4}$ | $60.8_{\pm5.8}$ | $99.5_{\pm0.2}$ |
| `halfcheetah-stitch-v0 - torso_height` | $33.3_{\pm9.9}$ | $72.6_{\pm7.2}$ | $87.1_{\pm7.7}$ | $71.5_{\pm10.7}$ | $80.1_{\pm6.8}$ | $96.9_{\pm1.4}$ |
| `halfcheetah-stitch-v0 - backfoot_height` | $25.0_{\pm3.8}$ | $28.6_{\pm2.7}$ | $30.0_{\pm6.3}$ | $28.0_{\pm3.4}$ | $27.3_{\pm3.9}$ | $47.0_{\pm2.4}$ |
| `halfcheetah-stitch-v0 - frontfoot_height` | $25.0_{\pm3.6}$ | $29.1_{\pm5.9}$ | $35.3_{\pm6.0}$ | $30.2_{\pm5.0}$ | $27.0_{\pm3.5}$ | $50.3_{\pm0.8}$ |
| `halfcheetah-stitch-v0 - all` | $30.0_{\pm6.8}$ | $52.1_{\pm7.6}$ | $58.9_{\pm11.3}$ | $48.4_{\pm12.5}$ | $57.4_{\pm4.7}$ | $\mathbf{78.0}_{\pm1.1}$ |
| `halfcheetah-vary-v0 - speed` | $33.3_{\pm6.9}$ | $63.3_{\pm15.5}$ | $56.4_{\pm23.2}$ | $54.3_{\pm14.3}$ | $37.8_{\pm5.8}$ | $96.7_{\pm0.1}$ |
| `halfcheetah-vary-v0 - angle` | $33.3_{\pm4.6}$ | $59.2_{\pm24.2}$ | $46.4_{\pm22.1}$ | $39.7_{\pm10.8}$ | $34.8_{\pm3.9}$ | $99.2_{\pm0.6}$ |
| `halfcheetah-vary-v0 - torso_height` | $33.3_{\pm7.6}$ | $79.3_{\pm10.9}$ | $92.6_{\pm7.5}$ | $77.0_{\pm11.8}$ | $36.2_{\pm6.1}$ | $98.8_{\pm0.3}$ |
| `halfcheetah-vary-v0 - backfoot_height` | $25.0_{\pm1.7}$ | $29.6_{\pm4.5}$ | $32.9_{\pm27.3}$ | $31.8_{\pm5.3}$ | $25.1_{\pm2.2}$ | $49.5_{\pm1.4}$ |
| `halfcheetah-vary-v0 - frontfoot_height` | $25.0_{\pm1.8}$ | $28.7_{\pm5.1}$ | $34.9_{\pm5.7}$ | $30.6_{\pm5.3}$ | $24.8_{\pm2.8}$ | $50.4_{\pm1.0}$ |
| `halfcheetah-vary-v0 - all` | $30.0_{\pm4.5}$ | $52.0_{\pm12.0}$ | $52.6_{\pm17.2}$ | $46.7_{\pm9.5}$ | $31.7_{\pm4.2}$ | $\mathbf{78.9}_{\pm0.7}$ |
| `humenv-simple-v0 - simple-head_height` | $50.0_{\pm44.4}$ | $89.1_{\pm22.0}$ | $79.2_{\pm26.7}$ | $79.4_{\pm26.9}$ | $\mathbf{99.6}_{\pm0.0}$ | $\mathbf{99.6}_{\pm0.0}$ |
| `humenv-simple-v0 - all` | $50.0_{\pm44.4}$ | $89.1_{\pm22.0}$ | $79.2_{\pm26.7}$ | $79.4_{\pm26.9}$ | $\mathbf{99.6}_{\pm0.0}$ | $\mathbf{99.6}_{\pm0.0}$ |
| `humenv-complex-v0 - complex-speed` | $33.3_{\pm5.2}$ | $32.6_{\pm7.1}$ | $32.1_{\pm13.6}$ | $34.3_{\pm4.7}$ | $34.1_{\pm5.8}$ | $\mathbf{83.7}_{\pm5.9}$ |
| `humenv-complex-v0 - complex-head_height` | $33.3_{\pm2.7}$ | $61.6_{\pm18.5}$ | $57.1_{\pm23.3}$ | $61.1_{\pm9.2}$ | $32.4_{\pm1.3}$ | $\mathbf{83.3}_{\pm6.6}$ |
| `humenv-complex-v0 - all` | $33.3_{\pm4.0}$ | $47.1_{\pm12.8}$ | $44.6_{\pm18.4}$ | $47.7_{\pm6.9}$ | $33.2_{\pm3.5}$ | $\mathbf{83.5}_{\pm6.2}$ |

**Style-conditioned task-performance optimization results (Table 4):** We see in Table 4 that choosing SORL's temperature $\beta_{\text{SORL}}$ is challenging, as finding a good balance between style alignment and task performance is highly sensitive to its value. For instance, in `halfcheetah-vary-v0 - speed`, as in many other settings, increasing $\beta_{\text{SORL}}$ from 0 to 1 leads to an immediate drop in style alignment. In `halfcheetah-vary-v0 - torso_height`, the decreases occur more gradually, with drops appearing both when moving from $\beta_{\text{SORL}} = 0$ to $\beta_{\text{SORL}} = 1$ and from $\beta_{\text{SORL}} = 1$ to $\beta_{\text{SORL}} = 3$. In contrast, SCIQL shows no such degradation. These examples highlight that tuning SORL's temperature for style-conditioned task-performance optimization can be troublesome, as it requires precise adjustment and the optimal value may vary across styles. SCIQL's temperature parameter $\beta_{\text{SCIQL}}$ differs fundamentally: it does not encode the trade-off between style alignment and task performance. Instead, it is inherited directly from IQL's temperature parameter $\beta_{\text{IQL}}$, while the trade-off itself is handled by the Gated Advantage Weighted Regression. Experimentally, we find that setting $\beta_{\text{SCIQL}}$ equal to the values of $\beta_{\text{IQL}}$ commonly used in the literature, typically chosen as 1.0, 3.0, and 10.0 (Kostrikov et al., 2022; Park et al., 2023; 2025), is an effective heuristic. Hence, SCIQL maintains strong alignment by design while significantly improving task performance, without requiring precise fine-tuning.

*Table 4.* **Style-conditioned task-performance optimization results (full).**

| Dataset | Metric | SORL ($\beta = 0$) | SORL ($\beta = 1$) | SORL ($\beta = 3$) | SCIQL ($\lambda$) | SCIQL ($\lambda > r$) | SCIQL ($r > \lambda$) |
|---|---|---|---|---|---|---|---|
| circle2d-inplace-v0 - all | Style | $58.9 \pm 2.7$ | $54.5 \pm 4.6$ | $53.9 \pm 4.2$ | $74.6 \pm 9.3$ | $71.6 \pm 4.8$ | $47.9 \pm 9.3$ |
| circle2d-inplace-v0 - all | Task | $16.6 \pm 6.2$ | $70.4 \pm 3.8$ | $73.6 \pm 3.3$ | $6.6 \pm 2.8$ | $68.6 \pm 6.9$ | $89.1 \pm 3.3$ |
| circle2d-inplace-v0 - position | Style | $14.9 \pm 11.6$ | $15.5 \pm 5.5$ | $12.1 \pm 3.2$ | $98.0 \pm 0.3$ | $96.1 \pm 1.9$ | $31.5 \pm 6.8$ |
| circle2d-inplace-v0 - position | Task | $12.8 \pm 7.4$ | $79.2 \pm 8.8$ | $80.4 \pm 7.7$ | $2.6 \pm 0.6$ | $17.3 \pm 4.1$ | $69.3 \pm 7.8$ |
| circle2d-inplace-v0 - movement_direction | Style | $5.3 \pm 4.2$ | $5.5 \pm 3.4$ | $4.7 \pm 1.7$ | $20.5 \pm 4.4$ | $14.5 \pm 2.3$ | $12.5 \pm 0.8$ |
| circle2d-inplace-v0 - movement_direction | Task | $0.5 \pm 0.1$ | $0.6 \pm 0.1$ | $0.6 \pm 0.2$ | $1.3 \pm 0.2$ | $80.8 \pm 11.3$ | $93.4 \pm 3.3$ |
| circle2d-inplace-v0 - turn_direction | Style | $100.0 \pm 0.1$ | $98.2 \pm 1.3$ | $97.9 \pm 2.2$ | $82.6 \pm 26.3$ | $85.5 \pm 11.3$ | $64.0 \pm 16.9$ |
| circle2d-inplace-v0 - turn_direction | Task | $14.3 \pm 3.2$ | $88.4 \pm 1.7$ | $90.1 \pm 3.1$ | $6.9 \pm 5.8$ | $90.8 \pm 3.7$ | $95.0 \pm 1.9$ |
| circle2d-inplace-v0 - radius | Style | $99.8 \pm 0.4$ | $77.1 \pm 12.2$ | $72.6 \pm 5.3$ | $96.1 \pm 1.3$ | $99.9 \pm 0.1$ | $57.1 \pm 16.3$ |
| circle2d-inplace-v0 - radius | Task | $28.3 \pm 10.0$ | $78.0 \pm 4.6$ | $87.4 \pm 2.3$ | $6.5 \pm 3.2$ | $53.9 \pm 10.4$ | $90.2 \pm 2.2$ |
| circle2d-inplace-v0 - speed | Style | $99.9 \pm 0.0$ | $97.4 \pm 4.8$ | $96.2 \pm 5.0$ | $91.6 \pm 13.3$ | $94.5 \pm 7.6$ | $88.4 \pm 14.7$ |
| circle2d-inplace-v0 - speed | Task | $21.0 \pm 8.2$ | $86.3 \pm 3.6$ | $91.8 \pm 2.4$ | $19.5 \pm 6.2$ | $91.5 \pm 2.1$ | $93.2 \pm 2.0$ |
| circle2d-inplace-v0 - curvature_noise | Style | $33.3 \pm 0.0$ | $33.5 \pm 0.3$ | $39.8 \pm 8.0$ | $59.1 \pm 6.1$ | $38.9 \pm 5.5$ | $33.6 \pm 0.3$ |
| circle2d-inplace-v0 - curvature_noise | Task | $22.8 \pm 8.0$ | $89.6 \pm 4.2$ | $91.3 \pm 4.2$ | $2.6 \pm 0.8$ | $77.5 \pm 9.7$ | $93.3 \pm 2.4$ |
| circle2d-navigate-v0 - all | Style | $60.0 \pm 3.3$ | $58.0 \pm 5.2$ | $57.6 \pm 4.0$ | $75.5 \pm 4.7$ | $76.5 \pm 2.9$ | $56.7 \pm 6.1$ |
| circle2d-navigate-v0 - all | Task | $18.5 \pm 7.3$ | $69.7 \pm 3.9$ | $72.7 \pm 3.9$ | $7.9 \pm 4.6$ | $66.2 \pm 6.5$ | $87.7 \pm 3.3$ |
| circle2d-navigate-v0 - position | Style | $22.3 \pm 14.8$ | $15.7 \pm 4.5$ | $13.9 \pm 3.1$ | $98.4 \pm 0.2$ | $96.0 \pm 2.2$ | $35.9 \pm 10.4$ |
| circle2d-navigate-v0 - position | Task | $19.8 \pm 10.2$ | $63.3 \pm 13.8$ | $69.4 \pm 13.1$ | $2.8 \pm 0.6$ | $20.1 \pm 2.8$ | $64.1 \pm 9.3$ |
| circle2d-navigate-v0 - movement_direction | Style | $4.9 \pm 3.7$ | $5.8 \pm 5.4$ | $5.6 \pm 4.1$ | $27.0 \pm 5.7$ | $18.4 \pm 4.0$ | $12.6 \pm 0.8$ |
| circle2d-navigate-v0 - movement_direction | Task | $0.4 \pm 0.0$ | $0.7 \pm 0.6$ | $0.4 \pm 0.1$ | $1.1 \pm 0.1$ | $63.3 \pm 13.4$ | $94.5 \pm 1.3$ |
| circle2d-navigate-v0 - turn_direction | Style | $100.0 \pm 0.1$ | $99.6 \pm 0.4$ | $99.8 \pm 0.1$ | $96.0 \pm 5.7$ | $100.0 \pm 0.0$ | $81.9 \pm 6.3$ |
| circle2d-navigate-v0 - turn_direction | Task | $18.4 \pm 11.4$ | $92.5 \pm 3.2$ | $93.4 \pm 2.6$ | $2.7 \pm 1.3$ | $94.4 \pm 2.4$ | $95.4 \pm 1.4$ |
| circle2d-navigate-v0 - radius | Style | $99.7 \pm 0.4$ | $91.2 \pm 7.0$ | $91.3 \pm 11.5$ | $95.8 \pm 5.6$ | $99.7 \pm 0.1$ | $77.1 \pm 16.8$ |
| circle2d-navigate-v0 - radius | Task | $30.9 \pm 9.4$ | $83.0 \pm 2.8$ | $88.0 \pm 1.8$ | $16.3 \pm 7.4$ | $64.3 \pm 8.4$ | $87.1 \pm 3.8$ |
| circle2d-navigate-v0 - speed | Style | $99.6 \pm 0.7$ | $97.1 \pm 6.3$ | $99.6 \pm 0.8$ | $96.0 \pm 4.5$ | $99.2 \pm 1.1$ | $99.0 \pm 1.8$ |
| circle2d-navigate-v0 - speed | Task | $21.6 \pm 5.0$ | $89.8 \pm 3.4$ | $90.6 \pm 3.4$ | $15.3 \pm 8.7$ | $92.7 \pm 4.5$ | $95.3 \pm 2.2$ |
| circle2d-navigate-v0 - curvature_noise | Style | $33.3 \pm 0.0$ | $38.9 \pm 7.9$ | $35.4 \pm 4.6$ | $40.0 \pm 6.7$ | $45.8 \pm 9.8$ | $33.6 \pm 0.7$ |
| circle2d-navigate-v0 - curvature_noise | Task | $19.7 \pm 7.7$ | $88.8 \pm 3.6$ | $94.5 \pm 2.1$ | $9.0 \pm 9.7$ | $62.4 \pm 7.5$ | $89.9 \pm 4.7$ |
| halfcheetah-fix-v0 - all | Style | $53.1 \pm 10.6$ | $44.4 \pm 6.1$ | $41.3 \pm 4.1$ | $78.0 \pm 1.8$ | $78.1 \pm 1.5$ | $49.7 \pm 5.4$ |
| halfcheetah-fix-v0 - all | Task | $32.1 \pm 8.4$ | $72.8 \pm 5.6$ | $80.6 \pm 3.1$ | $47.6 \pm 2.3$ | $56.5 \pm 2.5$ | $76.6 \pm 5.5$ |
| halfcheetah-fix-v0 - speed | Style | $73.0 \pm 20.3$ | $31.9 \pm 9.4$ | $34.6 \pm 2.2$ | $96.0 \pm 1.6$ | $95.6 \pm 3.1$ | $37.4 \pm 6.5$ |
| halfcheetah-fix-v0 - speed | Task | $42.5 \pm 13.2$ | $72.5 \pm 10.7$ | $84.1 \pm 2.4$ | $48.1 \pm 1.7$ | $51.6 \pm 1.9$ | $87.5 \pm 5.9$ |
| halfcheetah-fix-v0 - angle | Style | $60.0 \pm 15.5$ | $41.4 \pm 10.7$ | $30.9 \pm 2.7$ | $99.1 \pm 0.1$ | $99.5 \pm 0.1$ | $69.9 \pm 8.9$ |
| halfcheetah-fix-v0 - angle | Task | $26.2 \pm 5.3$ | $68.4 \pm 9.9$ | $83.2 \pm 4.2$ | $38.0 \pm 2.0$ | $48.9 \pm 1.9$ | $68.0 \pm 6.3$ |
| halfcheetah-fix-v0 - torso_height | Style | $73.2 \pm 8.9$ | $89.7 \pm 4.7$ | $84.0 \pm 7.9$ | $96.8 \pm 3.5$ | $98.0 \pm 1.9$ | $63.8 \pm 5.1$ |
| halfcheetah-fix-v0 - torso_height | Task | $33.8 \pm 8.9$ | $73.1 \pm 1.4$ | $73.9 \pm 1.7$ | $50.3 \pm 1.2$ | $51.5 \pm 1.0$ | $68.8 \pm 6.2$ |
| halfcheetah-fix-v0 - backfoot_height | Style | $28.4 \pm 2.8$ | $34.7 \pm 3.4$ | $31.0 \pm 4.6$ | $47.5 \pm 2.0$ | $49.2 \pm 1.2$ | $37.6 \pm 2.8$ |
| halfcheetah-fix-v0 - backfoot_height | Task | $34.7 \pm 6.6$ | $85.4 \pm 1.5$ | $86.4 \pm 1.9$ | $63.1 \pm 5.0$ | $76.2 \pm 1.6$ | $82.3 \pm 4.4$ |
| halfcheetah-fix-v0 - frontfoot_height | Style | $30.7 \pm 5.7$ | $24.1 \pm 2.4$ | $26.0 \pm 3.0$ | $50.5 \pm 0.8$ | $48.2 \pm 1.2$ | $39.9 \pm 3.8$ |
| halfcheetah-fix-v0 - frontfoot_height | Task | $23.5 \pm 7.9$ | $64.4 \pm 4.6$ | $75.4 \pm 5.3$ | $38.3 \pm 1.7$ | $54.5 \pm 5.9$ | $76.3 \pm 4.9$ |
| halfcheetah-stitch-v0 - all | Style | $48.4 \pm 12.5$ | $41.1 \pm 4.8$ | $42.1 \pm 4.9$ | $78.0 \pm 1.1$ | $60.8 \pm 6.0$ | $33.8 \pm 6.2$ |
| halfcheetah-stitch-v0 - all | Task | $31.9 \pm 10.3$ | $81.3 \pm 3.1$ | $78.3 \pm 5.6$ | $47.0 \pm 2.3$ | $70.0 \pm 6.0$ | $80.4 \pm 9.0$ |
| halfcheetah-stitch-v0 - speed | Style | $57.1 \pm 23.2$ | $34.0 \pm 2.3$ | $38.1 \pm 4.7$ | $96.3 \pm 0.5$ | $47.6 \pm 11.2$ | $32.6 \pm 5.2$ |
| halfcheetah-stitch-v0 - speed | Task | $32.7 \pm 14.3$ | $83.3 \pm 3.0$ | $81.3 \pm 5.0$ | $47.2 \pm 0.7$ | $78.7 \pm 8.5$ | $84.0 \pm 8.5$ |
| halfcheetah-stitch-v0 - angle | Style | $55.0 \pm 20.4$ | $31.5 \pm 3.3$ | $34.7 \pm 6.5$ | $99.5 \pm 0.2$ | $92.5 \pm 6.1$ | $38.0 \pm 6.0$ |
| halfcheetah-stitch-v0 - angle | Task | $25.5 \pm 8.8$ | $83.4 \pm 4.2$ | $79.7 \pm 9.7$ | $41.1 \pm 4.2$ | $54.8 \pm 6.6$ | $79.7 \pm 7.1$ |
| halfcheetah-stitch-v0 - torso_height | Style | $71.5 \pm 10.7$ | $83.0 \pm 16.0$ | $77.7 \pm 5.9$ | $96.9 \pm 1.4$ | $85.1 \pm 7.4$ | $44.5 \pm 8.3$ |
| halfcheetah-stitch-v0 - torso_height | Task | $33.7 \pm 10.9$ | $74.1 \pm 1.3$ | $69.8 \pm 4.1$ | $48.3 \pm 2.2$ | $59.5 \pm 5.5$ | $82.1 \pm 7.5$ |
| halfcheetah-stitch-v0 - backfoot_height | Style | $28.0 \pm 3.4$ | $30.6 \pm 5.0$ | $32.0 \pm 3.7$ | $47.0 \pm 2.4$ | $39.1 \pm 3.8$ | $29.0 \pm 6.3$ |
| halfcheetah-stitch-v0 - backfoot_height | Task | $41.2 \pm 9.2$ | $87.0 \pm 1.4$ | $84.6 \pm 4.5$ | $60.7 \pm 3.7$ | $80.8 \pm 6.4$ | $76.2 \pm 9.8$ |
| halfcheetah-stitch-v0 - frontfoot_height | Style | $30.2 \pm 5.0$ | $26.5 \pm 2.9$ | $28.0 \pm 3.6$ | $50.3 \pm 0.8$ | $39.5 \pm 1.3$ | $24.8 \pm 5.0$ |
| halfcheetah-stitch-v0 - frontfoot_height | Task | $26.5 \pm 8.3$ | $78.5 \pm 5.3$ | $76.1 \pm 4.9$ | $37.8 \pm 0.8$ | $76.3 \pm 3.2$ | $79.8 \pm 12.0$ |
| halfcheetah-vary-v0 - all | Style | $46.7 \pm 9.5$ | $37.0 \pm 3.0$ | $31.1 \pm 2.0$ | $78.9 \pm 0.7$ | $77.8 \pm 1.0$ | $41.8 \pm 5.0$ |
| halfcheetah-vary-v0 - all | Task | $35.9 \pm 9.0$ | $79.0 \pm 3.2$ | $82.6 \pm 3.1$ | $50.6 \pm 1.3$ | $58.0 \pm 1.7$ | $84.6 \pm 3.2$ |
| halfcheetah-vary-v0 - speed | Style | $54.3 \pm 14.3$ | $33.3 \pm 0.3$ | $33.4 \pm 0.2$ | $96.7 \pm 0.1$ | $96.9 \pm 0.4$ | $40.7 \pm 6.1$ |
| halfcheetah-vary-v0 - speed | Task | $42.7 \pm 9.3$ | $88.2 \pm 2.4$ | $88.7 \pm 2.2$ | $48.1 \pm 1.3$ | $50.7 \pm 0.9$ | $84.1 \pm 5.2$ |
| halfcheetah-vary-v0 - angle | Style | $39.7 \pm 10.8$ | $32.9 \pm 4.2$ | $31.8 \pm 2.0$ | $99.2 \pm 0.6$ | $98.7 \pm 1.8$ | $44.3 \pm 5.2$ |
| halfcheetah-vary-v0 - angle | Task | $19.0 \pm 7.4$ | $83.1 \pm 3.6$ | $84.7 \pm 2.3$ | $48.0 \pm 2.1$ | $55.3 \pm 1.1$ | $84.8 \pm 3.0$ |
| halfcheetah-vary-v0 - torso_height | Style | $77.0 \pm 11.8$ | $60.7 \pm 4.1$ | $36.9 \pm 3.2$ | $98.8 \pm 0.3$ | $98.8 \pm 0.3$ | $59.3 \pm 7.1$ |
| halfcheetah-vary-v0 - torso_height | Task | $37.3 \pm 11.7$ | $68.2 \pm 2.9$ | $74.0 \pm 3.0$ | $50.5 \pm 0.5$ | $50.9 \pm 1.3$ | $87.2 \pm 1.9$ |
| halfcheetah-vary-v0 - backfoot_height | Style | $31.8 \pm 5.3$ | $32.8 \pm 3.8$ | $27.4 \pm 3.5$ | $49.5 \pm 1.4$ | $45.7 \pm 1.2$ | $28.2 \pm 2.9$ |
| halfcheetah-vary-v0 - backfoot_height | Task | $48.1 \pm 7.5$ | $80.3 \pm 2.9$ | $82.6 \pm 4.7$ | $69.0 \pm 1.7$ | $75.0 \pm 1.8$ | $87.9 \pm 1.6$ |
| halfcheetah-vary-v0 - frontfoot_height | Style | $30.6 \pm 5.3$ | $25.4 \pm 2.8$ | $25.9 \pm 1.3$ | $50.4 \pm 1.0$ | $48.7 \pm 1.2$ | $36.5 \pm 3.6$ |
| halfcheetah-vary-v0 - frontfoot_height | Task | $32.4 \pm 8.9$ | $75.4 \pm 4.0$ | $83.0 \pm 3.1$ | $37.5 \pm 1.1$ | $58.0 \pm 3.2$ | $79.0 \pm 4.3$ |
| humenv-simple-v0 - simple-head_height | Style | $79.4 \pm 26.9$ | $99.1 \pm 0.9$ | $99.4 \pm 0.4$ | $99.6 \pm 0.0$ | $99.6 \pm 0.1$ | $99.5 \pm 0.2$ |
| humenv-simple-v0 - simple-head_height | Task | $14.6 \pm 14.5$ | $16.0 \pm 7.5$ | $20.0 \pm 12.5$ | $19.1 \pm 7.1$ | $31.7 \pm 4.8$ | $36.5 \pm 0.4$ |
| humenv-simple-v0 - all | Style | $79.4 \pm 26.9$ | $99.1 \pm 0.9$ | $99.4 \pm 0.4$ | $99.6 \pm 0.0$ | $99.6 \pm 0.1$ | $99.5 \pm 0.2$ |
| humenv-simple-v0 - all | Task | $14.6 \pm 14.5$ | $16.0 \pm 7.5$ | $20.0 \pm 12.5$ | $19.1 \pm 7.1$ | $31.7 \pm 4.8$ | $36.5 \pm 0.4$ |
| humenv-complex-v0 - complex-speed | Style | $34.3 \pm 4.7$ | $28.8 \pm 8.4$ | $22.6 \pm 11.1$ | $83.7 \pm 5.9$ | $91.6 \pm 8.9$ | $33.3 \pm 3.7$ |
| humenv-complex-v0 - complex-speed | Task | $5.7 \pm 1.6$ | $39.7 \pm 5.2$ | $33.6 \pm 8.2$ | $12.0 \pm 1.6$ | $16.2 \pm 2.3$ | $40.0 \pm 2.6$ |
| humenv-complex-v0 - complex-head_height | Style | $61.1 \pm 9.2$ | $22.0 \pm 13.6$ | $24.3 \pm 18.9$ | $83.3 \pm 6.6$ | $90.1 \pm 9.3$ | $33.3 \pm 4.9$ |
| humenv-complex-v0 - complex-head_height | Task | $4.5 \pm 3.8$ | $19.6 \pm 5.3$ | $20.5 \pm 9.3$ | $10.0 \pm 2.4$ | $15.7 \pm 2.6$ | $41.9 \pm 3.8$ |
| humenv-complex-v0 - all | Style | $47.7 \pm 6.9$ | $25.4 \pm 11.0$ | $23.5 \pm 15.0$ | $83.5 \pm 6.2$ | $90.8 \pm 9.1$ | $33.3 \pm 4.3$ |
| humenv-complex-v0 - all | Task | $5.1 \pm 2.7$ | $29.7 \pm 5.2$ | $27.1 \pm 8.8$ | $11.0 \pm 2.2$ | $15.9 \pm 2.5$ | $41.0 \pm 3.2$ |

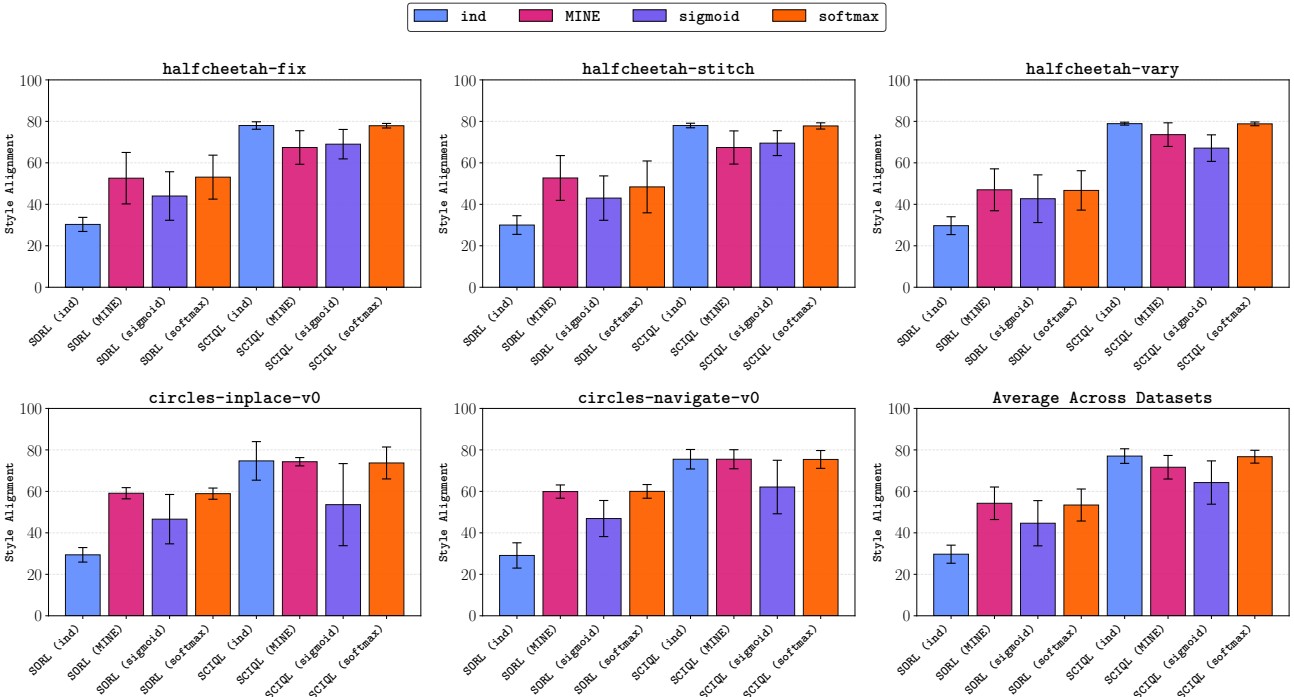

*Figure 17.* **Style alignment histograms for different** $p_{\pi_{\mathcal{D}}}^{\lambda}(z \mid s, a)$ **estimation strategies.**

## E. Ablations

For the following, we use the different distributions defined in Appendix C to answer the following questions:

1. Appendix E.1: How should we estimate $p_{\pi_{\mathcal{D}}}^{\lambda}(z \mid s, a)$?

2. Appendix E.2: What is the impact of the choice of $p_{\mathrm{m}}^{\lambda(\mathcal{D})}$?

3. Appendix E.3: How robust is SCIQL to imperfect style annotations?

### E.1. How should we estimate $p_{\pi_{\mathcal{D}}}^{\lambda}(z \mid s, a)$?

Estimating $p_{\pi_{\mathcal{D}}}^{\lambda}(z \mid s, a)$ relates to estimating the correspondence between a state-action pair and a style which is a key component of our problem. We tested for this purpose four distinct strategies to form an estimator $\chi(s, a, z)$ of $p_{\pi_{\mathcal{D}}}^{\lambda}(z \mid s, a)$. A first strategy noted **ind** consists in taking as the estimator the indicator of $z^{\lambda}$, the associated label of $(s, a, s')$ within $\lambda(\mathcal{D})$:

$$\forall (s, a, z), \chi_{\mathrm{ind}}(s, a, z) = \chi_{\mathrm{ind}}(z^{\lambda}, z) = \mathbb{1}(z = z^{\lambda}) \tag{55}$$

As $\lambda$ can attribute several labels to $(s, a)$ within $\mathcal{D}$, we can state that:

$$\forall z \in \mathcal{L}(\lambda), \forall (s, a) \in \mathcal{D}, \mathbb{E}_{z^{\lambda} \sim p_{\mathrm{c}}^{\lambda(\mathcal{D})}(\cdot \mid s, a)}[\chi_{\mathrm{ind}}(z^{\lambda}, z)] = \mathbb{E}_{z^{\lambda} \sim p_{\mathrm{c}}^{\lambda(\mathcal{D})}(\cdot \mid s, a)}[\mathbb{1}(z = z^{\lambda})] \approx p_{\pi_{\mathcal{D}}}^{\lambda}(z \mid s, a) \tag{56}$$

as the expectation of an indicator variable is the probability of its associated event. Hence, using $\chi_{\mathrm{ind}}$ can be justified when relying on a sufficient number of samples during training.

Another approach noted **MINE** is to use the MINE estimator described in Appendix C to estimate:

$$T_{\pi_{\mathcal{D}}}^{\lambda,*}(s, a, z) = \log \frac{p_{\pi_{\mathcal{D}}}^{\lambda}(s, a, z)}{p_{\pi_{\mathcal{D}}}^{\lambda}(s, a) p_{\pi_{\mathcal{D}}}^{\lambda}(z)} = \log \frac{p_{\pi_{\mathcal{D}}}^{\lambda}(z \mid s, a)}{p_{\pi_{\mathcal{D}}}^{\lambda}(z)} \tag{57}$$

by optimizing:

$$J_{\mathrm{MINE}}(T_{\pi_{\mathcal{D}}}^{\lambda}) = \mathbb{E}_{(s,a) \sim p^{\mathcal{D}}(s,a), z \sim p_{\mathrm{c}}^{\lambda(\mathcal{D})}(z \mid s, a)}[T_{\pi_{\mathcal{D}}}^{\lambda}(s, a, z)] - \log \left( \mathbb{E}_{(s,a) \sim p^{\mathcal{D}}(s,a), z \sim p_{\mathrm{r}}^{\mathcal{D}}(z)} \left[ e^{T_{\pi_{\mathcal{D}}}^{\lambda}(s,a,z)} \right] \right) \tag{58}$$

and taking:

$$\chi_{\text{MINE}}(s, a, z) = p_{\text{r}}^{\mathcal{D}}(z)e^{T_{\pi_{\mathcal{D}}}^{\lambda,*}(s,a,z)} \tag{59}$$

$$\approx p_{\text{r}}^{\mathcal{D}}(z)e^{\log \frac{p_{\pi_{\mathcal{D}}}^{\lambda}(z|s,a)}{p_{\pi_{\mathcal{D}}}^{\lambda}(z)}} \tag{60}$$

$$\approx p_{\text{r}}^{\mathcal{D}}(z)\frac{p_{\pi_{\mathcal{D}}}^{\lambda}(z \mid s, a)}{p_{\pi_{\mathcal{D}}}^{\lambda}(z)} \tag{61}$$

$$\approx p_{\pi_{\mathcal{D}}}^{\lambda}(z \mid s, a) \tag{62}$$

Also, as we seek to approximate $p_{\pi_{\mathcal{D}}}^{\lambda}(z \mid s, a) \in [0, 1]$ with discrete labels, we propose to train directly a neural network $\chi(s, a, z)$ within the MINE objective, taking $p_{\text{r}}^{\lambda(\mathcal{D})}(z)$ as an approximation of $p_{\pi_{\mathcal{D}}}^{\lambda}(z)$:

$$J_{\text{MINE}}(\chi) = \mathbb{E}_{(s,a)\sim p^{\mathcal{D}}(s,a),z\sim p_{\text{c}}^{\lambda(\mathcal{D})}(z|s,a)}[\log \frac{\chi(s, a, z)}{p_{\text{r}}^{\lambda(\mathcal{D})}(z)}] - \log \left( \mathbb{E}_{(s,a)\sim p^{\mathcal{D}}(s,a),z\sim p_{\text{r}}^{\lambda(\mathcal{D})}(z)} \left[ e^{\log \frac{\chi(s,a,z)}{p_{\text{r}}^{\lambda(\mathcal{D})}(z)}} \right] \right) \tag{63}$$

with $\chi$'s output activations taken as a sigmoid and a softmax to define the **sigmoid** and **softmax** strategies respectively. We evaluate the impact of each strategy on style alignment and report the results in Table 5 and Figure 17. For SORL, both **MINE** and **softmax** achieve the best performance, while for SCIQL the best results are obtained with **ind** and **softmax**. Accordingly, in our experiments we adopt **softmax** for SORL and **ind** for SCIQL.

*Table 5.* **Style alignments for different $p_{\pi_{\mathcal{D}}}^{\lambda}(z \mid s, a)$ estimation strategies.**

| Dataset | SORL (ind) | SORL (MINE) | SORL (sigmoid) | SORL (softmax) | SCIQL (ind) | SCIQL (MINE) | SCIQL (sigmoid) | SCIQL (softmax) |
|---|---|---|---|---|---|---|---|---|
| halfcheetah-fix-v0 | $30.3_{\pm3.4}$ | $52.6_{\pm12.4}$ | $44.0_{\pm11.7}$ | $53.1_{\pm10.6}$ | $78.0_{\pm1.8}$ | $67.4_{\pm8.1}$ | $69.0_{\pm7.1}$ | $77.9_{\pm1.1}$ |
| halfcheetah-stitch-v0 | $30.0_{\pm4.5}$ | $52.7_{\pm10.8}$ | $43.0_{\pm10.7}$ | $48.4_{\pm12.5}$ | $78.0_{\pm1.1}$ | $67.4_{\pm8.0}$ | $69.5_{\pm6.0}$ | $77.8_{\pm1.5}$ |
| halfcheetah-vary-v0 | $29.7_{\pm4.3}$ | $47.0_{\pm10.1}$ | $42.7_{\pm11.5}$ | $46.7_{\pm9.5}$ | $78.9_{\pm0.7}$ | $73.6_{\pm5.7}$ | $67.1_{\pm6.4}$ | $78.8_{\pm0.9}$ |
| circles2d-inplace-v0 | $29.4_{\pm3.5}$ | $59.1_{\pm2.7}$ | $46.6_{\pm11.9}$ | $58.9_{\pm2.7}$ | $74.7_{\pm9.3}$ | $74.3_{\pm2.0}$ | $53.6_{\pm19.8}$ | $73.7_{\pm7.7}$ |
| circles2d-navigate-v0 | $29.1_{\pm6.1}$ | $59.9_{\pm3.2}$ | $46.9_{\pm8.7}$ | $60.0_{\pm3.3}$ | $75.5_{\pm4.7}$ | $75.5_{\pm4.6}$ | $62.1_{\pm12.9}$ | $75.4_{\pm4.3}$ |
| all_datasets | $29.8_{\pm4.0}$ | $\mathbf{53.8}_{\pm8.6}$ | $44.9_{\pm11.3}$ | $\mathbf{53.2}_{\pm8.5}$ | $\mathbf{77.2}_{\pm3.2}$ | $70.9_{\pm6.0}$ | $64.9_{\pm10.1}$ | $\mathbf{76.9}_{\pm2.8}$ |

## E.2. What is the impact of the choice of $p_{\mathrm{m}}^{\lambda(\mathcal{D})}$?

To address the lower performance of SCIQL on the `turn_direction`, `radius`, and `speed` criteria of **Circle2d**, we evaluated SCIQL by sampling styles from $p_{\mathrm{c}}^{\lambda(\mathcal{D})}$ rather than $p_{\mathrm{r}}^{\lambda(\mathcal{D})}$. As shown in the histogram in Figure 18, using $p_{\mathrm{c}}^{\lambda(\mathcal{D})}$ improves style alignment to its maximum score, highlighting both SCIQL's flexibility in varying its style sampling distributions and the potential importance of this choice when optimizing style alignment.

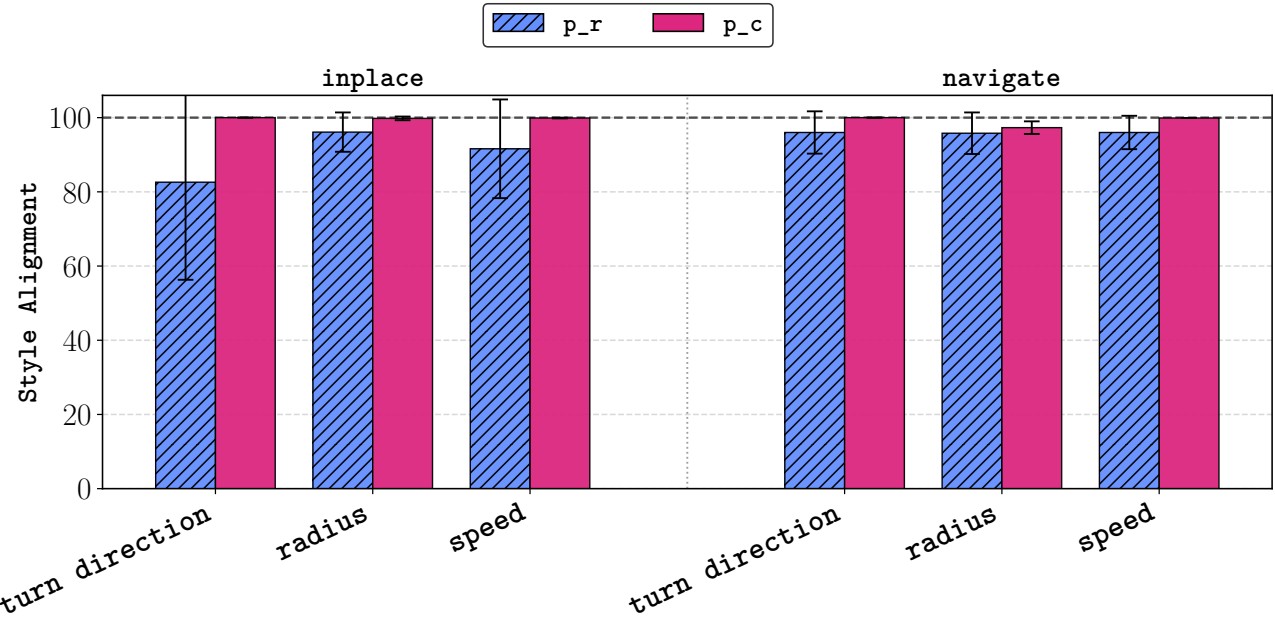

*Figure 18.* **SCIQL performance under $p_{\mathrm{r}}^{\lambda(\mathcal{D})}$ vs $p_{\mathrm{c}}^{\lambda(\mathcal{D})}$?**

### E.3. How robust is SCIQL to imperfect style annotations?

While relying on labeling functions allows for explainable and precise style annotations, style annotations could in practice be imperfect due to the noisiness of domain experts. For instance, alternative labeling approaches such as human generated labels or VLMs could provide noisy labels due to biases, stochasticity and unclear cuts between style transitions. All those imperfections can have a significant impact on style alignment. Hence, to measure the robustness of SCIQL in comparison to the baselines, we simulate labeling imperfections by modifying the labeling procedure such that for a given criterion $\lambda$, each state-action-style triplet $(s_t, a_t, z_t^\lambda)$ of $\lambda(\mathcal{D})$ is polluted with a probability $\zeta$ by changing its label $z_t^\lambda$ to another label $\tilde{z}_t^\lambda$ sampled uniformly among other available labels of $\mathcal{L}(\lambda)$. We plot in Figure 19 the evolution of the style alignment of the different baselines for the `halfcheetah-fix-v0 - speed` tasks, `halfcheetah-fix-v0 - angle` tasks and the average of those evolutions as `halfcheetah-fix-v0 - speed + angle`.

First, for noise levels going from $0.0$ to $0.6$, we see that SCIQL maintains a very good style alignment. More precisely, SCIQL is on average (i.e. in `speed_label + angle_label`) better aligned with a noise level of $0.6$ than all of the other baselines with no noise. The other baselines lose all their alignment even for small noise levels such as $0.2$, obtaining style alignments equal to BC's, which means that the baselines consider any noisy label as uninformative noise and ignore them, losing all conditioning capabilities. This shows that **SCIQL is significantly more robust to label noise than any test baseline**, highlighting the benefits of integrating RL signals to style alignment training.

Second, above a certain noise threshold $\bar{\zeta}$, we see that SCIQL's alignment plummets toward 0, which is in fact a good feature. A possible intuition is that this threshold corresponds to the noise level above which the true labeling of each state-action pair is no longer the majority in the noisy dataset. Beyond this threshold, for SCIQL, **the best outcome for alignment is to reach wrong labels**. Indeed, for each state-action pair $(s, a)$, the probability of labeling to the right label $z$ is $p_{\text{right}} = 1 - \zeta$, while the probability of choosing a wrong label is $p_{\text{wrong}} = \zeta$. Since wrong labels are sampled uniformly, each individual wrong label $\tilde{z}_i \in Z_{\text{wrong}} = \mathcal{L}(\lambda) \backslash \{z\}$ has a probability $p_i = \frac{\zeta}{|\lambda|-1}$ to be selected, $|\lambda|$ being the total number of labels in $\mathcal{L}(\lambda)$. Consequently, for the right label to maintain the majority position, the threshold needs to verify:

$$\forall \tilde{z}_i \in Z_{\text{wrong}}, p_{\text{true}} > p_i \Leftrightarrow p_{\text{true}} > \max_{\tilde{z}_i \in Z_{\text{wrong}}} p_i \Leftrightarrow 1 - \zeta > \frac{\zeta}{|\lambda|-1} \Leftrightarrow \frac{|\lambda|-1}{|\lambda|} > \zeta \tag{64}$$

Also, as described in Appendix A, both speed and angle criteria have the same number of $|\lambda| = 3$ labels each and as such, for both labels $\bar{\zeta} = \frac{|\lambda|-1}{|\lambda|} = \frac{2}{3}$, which **corresponds to the observed threshold and consequently supports our intuition**.

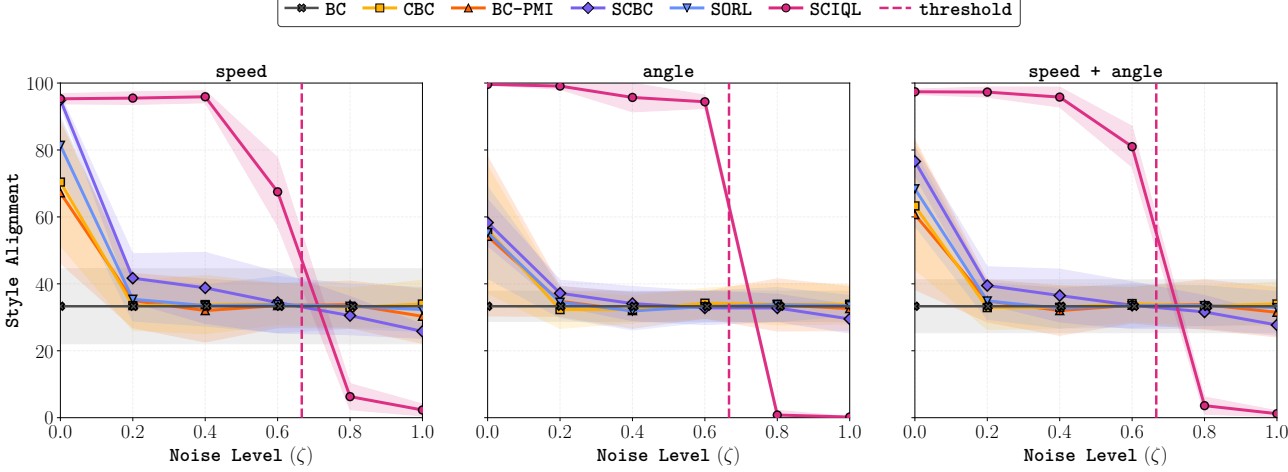

*Figure 19.* **Evolution of style alignment under noisy labels.** For noise labels $\zeta \in \{0.0, 0.2, ..., 1.0\}$, we compare the evolution of style alignment of **BC**, **CBC**, **BC-PMI**, **SCBC**, **SORL** and **SCIQL**. We see that SCIQL maintains an overall better alignment before the noise threshold where the true label is the majority, and then misaligns itself beyond the noise threshold, which corresponds to following intentionally the wrong styles accordingly to the noisy labeling.

