# OpenReview forum: "Offline Reinforcement Learning of High-Quality Behaviors Under Robust Style Alignment"
_ICML.cc/2026/Conference — ICML 2026 spotlight_

### Official Review · Reviewer_2NH6 · 2026-03-09

**Soundness:** 3
**Presentation:** 2
**Significance:** 3
**Originality:** 2
**Overall Recommendation:** 4
**Confidence:** 3

**Summary:**

This paper studies offline RL with style conditioning and aims to achieve robust style alignment and high task performance. This paper propose Style-Conditioned Implicit Q-Learning (SCIQL) and Gated advantage weighted regression to align style while optimizing task performance. This approach addresses challenges including distribution shift, credit assignment, and task-style misalignment. Experiments on Circle2d, HalfCheetah, and HumEnv demonstrate superior style alignment and task performance compared to baselines like SORL and BCPMI.

**Compliance With Llm Reviewing Policy:**

Affirmed.

**Final Justification:**

The rebuttal has addressed my major concerns. Therefore, I increase my score.

**Key Questions For Authors:**

See the weaknesses for addressing the main concerns.

**Limitations:**

yes

**Strengths And Weaknesses:**

**Strengths**
- The idea of learning a stylized and high-performing offline policy is novel and interesting.
- The experimental results demonstrate that SCIQL outperforms other compared stylized offline RL methods.

**Weaknesses & Questions**
- The paper highlights three challenges that SCIQL aims to address. However, the first challenge, distribution shift, is precisely the most common issue in offline RL and the use of IQL for addressing this issue is not an innovation of this article. In my opinion, the article spends too much space on this part, which reduces its clarity.
- The entire pipeline consists of three main steps: training the task advantage function, training the style advantage function, and Gated advantage weighted regression. This increases the time cost of the training.
- The method relies heavily on hand-crafted labeling functions. While interpretable, defining these functions for high-dimensional, unstructured tasks (e.g., raw pixels) is non-trivial. This raises the consideration of the scalability of the method.

---

> ### Author Rebuttal · Authors · 2026-03-30
>
> We thank **R-2NH6** for their review and constructive feedback. We appreciate their recognition of both the novelty of learning stylized, high-performing offline policies and the strong empirical performance of our method.
>
> ## On distribution shift
>
> > Q.1: The paper highlights three challenges that SCIQL aims to address. However, the first challenge, distribution shift, is precisely the most common issue in offline RL [...]. In my opinion, the article spends too much space on this part, which reduces its clarity.
>
> We thank **R-2NH6** for raising this point and fully understand their concern. As the goal of Section 4 is to present the design of our algorithm, we originally structured it into three stages, reflecting the three objectives of our method: (1) optimizing the task reward, (2) optimizing the style alignment, and (3) optimizing the task reward under style alignment. Nevertheless, while the first component (1) forms the foundation of our approach, we agree that it is more appropriately presented as preliminary material.
>
> Hence, to improve clarity and better align with the introduction, we will revise the manuscript by simply **swapping** the content of Section 4 related to IQL with that of Section 3 related to the Style Alignment Objective Definition, as the former is better suited for preliminary material, while the latter more directly reflects our core contributions stated in the introduction (“Challenge 1: Style definition”).
>
> This results in the following organization:
>
> 3. Preliminaries
>  - Paragraph “Markov decision processes.”
>     - Objective: Introduce MDPs.
>  - Paragraph “Offline reinforcement learning.”
>    - Objective: Introduce the RL problem, the offline RL problem, and the distribution-shift problem.
>  - Paragraph “Implicit Q-Learning (IQL).”
>    - Objective: Introduce IQL as preliminary material.
> 4. Style-Conditioned Implicit Q-Learning
>   - 4.1 Addressing Challenge 1: Style definition
>     - Objective: Introduce a style definition and style alignment objective.
>   - 4.2 Addressing Challenge 2: State-style distribution shift
>     - Objective: Introduce SCIQL for style alignment.
>   - 4.3 Addressing Challenge 3: Task optimization under style alignment
>     - Objective: Introduce GAWR for task optimization under style alignment.
> 5. Experiments
>
> We thank the reviewer again for their insightful feedback. We hope this revision addresses their concern, and we would be happy to provide further clarification if needed.
>
> ## On training cost
>
> > Q.2: The entire pipeline consists of three main steps [...]. This increases the time cost of the training.
>
> As our method builds upon IQL, it inherits its sequential training structure, as originally described in the IQL paper [1]. However, while these steps can be presented sequentially, they are typically executed jointly within a single training loop in most IQL-based implementations [1,2,3].
>
> Indeed, since each pair of value functions (Q and V) is independent, and the gated advantage-weighted regression has a significantly lower computational cost than value function training, we perform all SCIQL training steps jointly within a single loop. This implementation choice, as noted in the rightmost column of l. 293, reduces both algorithmic complexity and overall training time.
>
> Moreover, the independence between value function pairs makes this structure particularly advantageous. For example, value function training can be parallelized across multiple GPUs to further accelerate computation. Additionally, trained value functions can be stored and reused, for instance, when modifying either the target task or the target style independently.
>
> ## On labeling functions
>
> > Q.3: The method relies heavily on hand-crafted labeling functions. [...] This raises the consideration of the scalability of the method.
>
> We agree with **R-2NH6** that the scalability of labeling is an important aspect, especially in high-dimensional settings. We emphasize that the way labels are obtained is a complementary research problem, as SCIQL itself is agnostic to the label source. While hand-crafted labeling functions are interpretable and suitable for controlled evaluation, alternative sources such as human annotators, LLMs/VLMs, or learned classifiers can be readily integrated to capture more abstract styles. A key consideration is that such labels may be noisy. However, SCIQL is inherently robust to label noise, as demonstrated in Appendix E.3. We refer the reviewer to our response to **R-cGKm** (“On labeling functions”) for further details.
>
> We hope that our concise responses, given the space constraints, satisfactorily address your concerns, and we remain available for further discussion.
>
> [1] Kostrikov & al., Offline Reinforcement Learning with Implicit Q-Learning, ICLR 2022.
>
> [2] Park & al., HIQL: Offline Goal-Conditioned RL with Latent States as Actions, Neurips 2023.
>
> [3] Park & al., OGBench: Benchmarking Offline Goal-Conditioned RL, ICLR 2025.

---

> > ### Author Rebuttal · Reviewer_2NH6 · 2026-04-03
> >
> > Thank the authors for addressing my concerns. I increase my score to 4.

---

> > > ### Author Response · Authors · 2026-04-07
> > >
> > > We are grateful to **R-2NH6** for their thoughtful comments, constructive feedback, and careful reading of our paper. We are pleased that our rebuttal helped clarify the points they raised and address the remaining concerns.

---

### Official Review · Reviewer_UncK · 2026-03-13

**Soundness:** 3
**Presentation:** 4
**Significance:** 3
**Originality:** 3
**Overall Recommendation:** 5
**Confidence:** 4

**Summary:**

This paper investigates the problem of style-conditioned offline reinforcement learning, where the goal is to learn a policy that maximizes task rewards while adhering to a specific behavioral style from a heterogeneous dataset. The authors identify a "style-reward conflict" in current offline RL methods and propose **SCIQL (Style-Conditioned Implicit Q-Learning)** to address it.

The framework is built on two key components: **Subtrajectory Labeling**, which defines style through local temporal windows (effectively a temporal discretization of behavior), and a **Gated Advantage Weighted Regression (GAWR)** mechanism. GAWR allows the agent to filter transitions that do not align with the target style and "stitch" together high-reward segments from various trajectories. The authors demonstrate that SCIQL outperforms several offline RL baselines in terms of both task performance and style alignment across various simulated environments.

**Compliance With Llm Reviewing Policy:**

Affirmed.

**Final Justification:**

The paper is technically sound and addresses a significant challenge in offline RL: balancing style alignment with task performance.

The proposed GAWR mechanism is an original and effective engineering solution for behavior stitching, and the empirical results across various locomotion tasks are compelling.

​The authors' rebuttal successfully addressed my primary concerns:
* Contextual Positioning: The authors clarified how their subtrajectory labeling complements prior work on state-space discretization and committed to integrating these perspectives.
* ​Methodological Robustness: Their explanation regarding advantage signals and adjustable window sizes w provides sufficient justification for handling state volatility.
* ​Limitations & Impact: I appreciate the commitment to explicitly discuss potential risks like behavioral forgery and labeling bias in the revised manuscript.

​While the dependency on manual labeling remains a constraint, SCIQL provides a robust framework for controllable AI. Given the technical quality and the authors' constructive engagement, I have increased my score to reflect a positive recommendation for acceptance.

**Key Questions For Authors:**

1. **Subtrajectory vs. State-Space Discretization:** The proposed subtrajectory window $w$ can be viewed as a form of temporal discretization for behavior. How does this temporal approach reconcile with methods that use state-space discretization (e.g., VQ-VAE based discrete states in **Lin et al., TMLR 2024**) to identify comparable decision-making points? Could incorporating state-level similarity help SCIQL better ground its style labels in intrinsic policy behavior rather than relying solely on extrinsic temporal windows?
2. **Assumption of Temporal Smoothness:** The labeling function $\lambda$ implicitly assumes that behavioral style is locally stationary within the window $\pm w$. In highly dynamic or high-frequency control environments where state volatility is high, how robust is this temporal windowing? Would the style alignment accuracy degrade if the state changes drastically within short intervals, and how might a more "perceptual" state-based distance metric mitigate this?
3. **Interpretability of the Learned Style:** In SCIQL, the style is a target condition (goal), but the underlying decision-making logic remains a black box. Can the authors provide more insight into *what* the policy actually learns to change—for instance, by analyzing the policy divergence at common states across different style conditions? This would bridge the gap between "satisfying a label" and "interpreting a decision-making style."

* Lin et al. Perceptual Similarity for Measuring Decision-Making Style and Policy Diversity in Games. TMLR 2024.

**Limitations:**

**No.** The authors do not adequately discuss the fundamental limitations or negative societal impacts. I suggest the following:

* **Heuristic Dependency:** The method relies heavily on human-defined labeling functions. The authors should discuss how the framework handles noisy or biased labels.
* **Temporal Smoothing Assumption:** The "subtrajectory window" assumes style remains stationary over time. This might fail in volatile environments where states/actions change rapidly.
* **Unsupervised Comparison:** The paper would benefit from discussing how this supervised approach compares to unsupervised style discovery (e.g., state-space discretization).
* **Societal Impact:** The "no content to highlight" statement is insufficient. The authors should address risks like behavioral forgery or the reinforcement of biases in offline data.

**Strengths And Weaknesses:**

## **Strengths**

1. **Effective Behavior Stitching:** The paper successfully applies the "stitching" property of offline RL to the style domain. SCIQL shows a robust ability to combine disparate fragments of behavioral data to satisfy complex constraints (style + goal), even when complete trajectories of a specific style/performance level are missing.
2. **Algorithmic Efficiency (GAWR):** The introduction of the GAWR mechanism is a practical contribution. By using a gated weight to prioritize style-consistent transitions, the method avoids the complex hyperparameter tuning often required by Lagrangian-based constrained optimization.
3. **Clarity and Empirical Breadth:** The paper is well-structured and easy to follow. The experimental results on various locomotion tasks (e.g., HalfCheetah, Humenv) provide strong evidence that the proposed method handles distribution shift more effectively than standard Behavior Cloning (BC) or existing style-conditioned methods.

## **Weaknesses**

1. **Heuristic Dependency on Labeling Functions:** The definition of style is entirely dependent on manually designed labeling functions $\lambda$. This creates a significant bottleneck; the "style" is only as good as the human's ability to define it. It lacks the capacity for unsupervised style discovery or identifying latent behavioral nuances.
2. **Limitations of Temporal Discretization:** By defining style via a fixed temporal window (subtrajectory), the method implicitly assumes temporal stationarity of behaviors. This approach is sensitive to the choice of window size $w$ and may fail in highly dynamic or volatile environments where the state-action distribution changes more rapidly than the window can capture.
3. **Lack of Decision-Making Interpretability:** The framework treats style as an extrinsic target label rather than an intrinsic policy characteristic. Unlike methods that analyze policy divergence at shared states (e.g., using state-space discretization), SCIQL does not explain *why* or *how* the decision-making logic changes between styles, potentially limiting its transparency in complex game AI or safety-critical applications.

---

> ### Author Rebuttal · Authors · 2026-03-30
>
> We thank **R-UncK** for their review and constructive feedback. We are pleased to see that the application of behavior stitching techniques, the efficiency of the GAWR as well as the clarity and empirical breadth of our paper is considered valuable.
>
> ## On labeling functions
>
> > **W.1: Heuristic Dependency on Labeling Functions, L.1: Heuristic Dependency, L.3: Unsupervised Comparison.**
>
> We agree with **R-UncK** that representation span and robustness to label noise are important considerations. Due to space constraints, we refer the reviewer to our response to **R-cGKm** (“On labeling functions”), which addresses both.
>
> Regarding unsupervised style discovery, the lack of guidance risks capturing arbitrary or uninformative behavioral variations, leading to latent “styles” that may not be semantically meaningful to users. It may also fail to capture relevant style variations, limiting user control.
>
> To ensure meaningful user control, we constrain the style space with simple labeling functions, reducing both spurious discoveries and omissions. That said, integrating discovery mechanisms while preserving style relevance is a promising direction.
>
> ## On temporal and state-space discretization
>
> > **Q.1: Subtrajectory vs. State-Space Discretization.**
>
> We thank **R-UncK** for pointing out **Lin et al. (2024)** [1], which introduces a playstyle similarity measure by extending the playstyle distance based on VQ-VAE state categories proposed in [2], incorporating multiscale discretization, a perceptual kernel, and intersection-over-union weighting. We view our work and [1] as complementary.
>
> On the one hand, the VQ-VAE categories in [1] can serve as learned perceptual decision-point abstractions of states, providing a practical way to generate semantically meaningful labels in complex observation spaces such as pixel inputs.
>
> On the other hand, SCIQL can address the disjunction between dataset categories highlighted in [1], as it can stitch and plan across disjoint state categories to achieve alignment.
>
> We will mention [1] and integrate this discussion in the revised manuscript’s related work and future work sections, as it indeed opens a very promising extension of our approach.
>
> ## On temporal windows and behavior volatility
>
> > **W.2: Limitations of Temporal Discretization, Q.2: Assumption of Temporal Smoothness, L.2: Temporal Smoothing Assumption.**
>
> We agree with **R-UncK** that high-volatility scenarios are a challenge for trajectory labeling via labeling functions. Our method presents several design choices to mitigate such risks:
>  - As noted in the rightmost column of l. 143, unlike [3], our framework defines labeling functions over time windows rather than full trajectories. This enables the use of oracle labelers that detect punctual style expressions on short sub-trajectories, making them more robust to state-level volatility. Moreover, by adjusting the window size to the nominal timescale of each style, our method can capture both short-term, high-volatility behaviors and longer-term, smoother patterns.
>  - As shown in Appendix E.1, SCIQL can incorporate classifier learning (e.g., $p(z|s,a)$ via MINE, as in [3]) without performance loss to account for uncertainty in $(s,a)$’s contribution to the style $z$.
>  - As demonstrated in Appendix E.3, the use of advantage signals makes SCIQL particularly robust to label noise.
>
> Additionally, future work could draw on [1] to design perceptual abstraction-based labelers and hierarchical temporal labeling, improving robustness to state-level volatility.
>
> ## On safety and interpretability
> > **W.3: Lack of Decision-Making Interpretability, Q.3: Interpretability of the Learned Style.**
>
> Regarding safety, SCIQL optimizes for rapid style alignment under discounting, even via temporary misalignment (e.g., crawling before running). We argue that this can improve robustness in game AI and safety-critical settings.
>
> Regarding explainability, SCIQL inherits limited interpretability from deep RL. While this subject is of high importance for deployment in critical applications, it lies outside the scope of this study. As such, improving the explainability of our policies, for instance by using the style similarity measure from [1], would be a very valuable direction for future work.
>
> ## On societal impact
>
> > **L.4: Societal Impact.**
>
> We will clarify this aspect in the revised manuscript by discussing potential misuse, including biased labels that may induce discrimination, behavioral forgery, and privacy concerns.
>
> We hope that our responses, given the space limits, address your concerns, and we remain available for further discussion.
>
> [1] Lin & al., Perceptual Similarity for Measuring Decision-Making Style and Policy Diversity in Games, TMLR 2024.
>
> [2] Lin & al., An unsupervised video game playstyle metric via state discretization, UAI 2021.
>
> [3] Yang, Yao & al., Diverse Policies Recovering via Pointwise Mutual Information Weighted Imitation Learning, ICLR 2025.

---

> > ### Author Rebuttal · Reviewer_UncK · 2026-04-02
> >
> > The authors have adequately addressed the concerns raised in the original review:
> >
> >
> > * Context and Literature: The authors have clarified the positioning of their work and committed to incorporating a more comprehensive discussion on behavioral similarity and prior metrics into the revised manuscript.
> > * ​Methodological Clarification: The rebuttal provides a sound explanation for the use of subtrajectory labeling and window-based discretization, particularly regarding its robustness to state volatility through the use of advantage signals.
> > * Limitations and Impact: The commitment to explicitly discuss potential negative societal impacts and the inherent limitations of supervised labeling resolves the previous omissions in the paper.
> >
> > ​The clarifications provided in the rebuttal are sufficient, and the proposed revisions will strengthen the final version of the paper. I have adjusted my score to reflect this.

---

> > > ### Author Response · Authors · 2026-04-07
> > >
> > > We sincerely appreciate **R-UncK**’s detailed evaluation, valuable suggestions, and engagement with our work. We are pleased that our responses helped clarify and address the main concerns they raised.

---

### Official Review · Reviewer_cGKm · 2026-03-13

**Soundness:** 4
**Presentation:** 4
**Significance:** 3
**Originality:** 3
**Overall Recommendation:** 5
**Confidence:** 4

**Summary:**

This paper presents a compelling shift from traditional reinforcement learning, which typically focuses on the singular goal of return maximization, toward a framework that prioritizes how a task is performed through behavioral styles. By introducing Style-Conditioned Implicit Q-Learning (SCIQL), the authors provide a fresh perspective on stylized policy learning from offline data.

The methodology effectively bridges Goal-Conditioned RL (GCRL) and Multi-Objective RL (MORL). While the authors formally cast style alignment as a generalization of GCRL—where the "goal" is to visit state-action pairs consistent with a target label—the inherent tension between style and task performance resonates strongly with multi-objective optimization challenges. This dual characteristic allows the framework to benefit from GCRL’s trajectory stitching capabilities while addressing the conflicting nature of diverse objectives.

A significant technical contribution is the Gated Advantage Weighted Regression (GAWR) mechanism. Rather than adopting a standard linear scalarization of rewards, which often causes one objective to diminish the other, GAWR manages these objectives separately through a gating function. By using the style advantage ($A^\lambda$) as a sigmoid-activated gate for the task advantage ($A^r$), the algorithm selectively incorporates task performance signals only when they are unlikely to jeopardize style alignment.

**Compliance With Llm Reviewing Policy:**

Affirmed.

**Final Justification:**

The authors have adequately addressed my questions. While encoding behavioral styles becomes increasingly challenging as the desired style grows more sophisticated, this work provides a simple yet effective approach to optimizing behavior under style constraints. Although the style constraints remain hand-crafted, the breadth of styles explored in the experiments demonstrates sufficient value as a strong baseline for future work in this direction. I maintain my original score.

**Key Questions For Authors:**

**Mapping and Encoding Complexity**

While I have no major concerns regarding the organization of the paper, I remain thoughtful about the practical challenges of mapping subtrajectories to behavioral styles. The primary difficulty lies in the fact that 'style' is inherently elusive and complex to encode directly.

I believe a compelling direction for future research would be to express these behavioral styles in natural language. If styles can be articulated in human terms, leveraging Large Language Models (LLMs) to encode these linguistic descriptions could bridge the gap between human-interpretable intent and machine-learned representations. This approach could potentially replace or augment the current hard-coded labeling functions with a more flexible, open-vocabulary style-conditioning space.

**Limitations:**

Yes

**Strengths And Weaknesses:**

**Strengths**

**Systematic Problem Solving**: The manuscript is exceptionally well-structured, explicitly identifying three core challenges—distribution shift, sparse/missing style supervision, and the intractability of style-optimal projections. Each challenge is systematically mapped to a specific technical solution within the SCIQL and GAWR frameworks, providing a clear and logical narrative.

**Robustness via Exponential Weighting**: A key highlight is the transition from simpler weighting schemes, like those in SORL, to an exponential advantage-weighted form. By placing the style advantage $A^\lambda$ in the exponent, the algorithm achieves significantly more robust style alignment. This is empirically validated in Section E.3, where SCIQL maintains high performance even under high levels of label noise that cause baseline methods to fail.

**Transparency and Visualization**: The accompanying GitHub project page provides excellent visualizations that bring the "styles" to life, making the qualitative improvements as clear as the quantitative ones. The inclusion of Pareto front analyses further clarifies the trade-offs between task performance and style alignment.

**Weakness**

There are no major technical or conceptual flaws identified in the current version of the manuscript.

---

> ### Author Rebuttal · Authors · 2026-03-30
>
> We thank **R-cGKm** for their review and appreciate their recognition of our manuscript’s clarity, our method’s performance, and the inclusion of visualizations.
>
> ## On labeling functions
>
> > Q.1: While I have no major concerns regarding the organization of the paper, I remain thoughtful about the practical challenges of mapping subtrajectories to behavioral styles. The primary difficulty lies in the fact that 'style' is inherently elusive and complex to encode directly.
> I believe a compelling direction for future research would be to express these behavioral styles in natural language. If styles can be articulated in human terms, leveraging Large Language Models (LLMs) to encode these linguistic descriptions could bridge the gap between human-interpretable intent and machine-learned representations. This approach could potentially replace or augment the current hard-coded labeling functions with a more flexible, open-vocabulary style-conditioning space.
>
> We agree with **R-cGKm**’s statement. While hand-crafted labeling functions offer clear advantages such as being accurate, cost-effective, and interpretable, enabling rigorous benchmarking, they are limited in their ability to capture more abstract behavioral styles.
>
> Fortunately, SCIQL is agnostic to the label source, making it straightforward to integrate labels from alternative sources, such as human annotators, LLMs/VLMs, or learned classifiers, to broaden the representation of style labels.
>
> In particular, we agree with **R-cGKm** that the use of LLMs is a very promising future research direction, as they could leverage their general human-like knowledge to automatically map trajectories to more abstract or conceptual styles without further human supervision.
>
> However, their use also introduces challenges, including high computational cost, susceptibility to noisy (i.e., occasionally incorrect) labels, and possible grounding issues for LLMs/VLMs within the RL environment. Nonetheless, we believe that our method provides a strong testbed for exploring such techniques:
>
>  - First, labeling costs can be reduced through semi-supervised approaches, where only a subset of the data is annotated to train a lightweight classifier.
>  - Second, label noise can be effectively handled by SCIQL: as shown in Appendix E.3, our method demonstrates strong robustness to noisy labels, which suggests promising scalability to more abstract and potentially noisier labeling schemes.
>  - Finally, prior work has explored grounding VLMs, for example, in [1], to construct zero-shot reward functions in RL settings, indicating encouraging potential for leveraging VLMs in subtrajectory style labeling.
>
> We hope this addresses your question, and we remain happy to provide any further clarification if needed.
>
> [1] Rocamonde & al., Vision-Language Models are Zero-Shot Reward Models for Reinforcement Learning, ICLR 2024.

---

> > ### Author Rebuttal · Reviewer_cGKm · 2026-04-02
> >
> > The author has addressed my questions.
> >
> > While the style becomes more complex to encode as the required style grows more sophisticated, I believe this work offers a simple yet effective method for optimizing behavior under style constraints. Although the style constraints in this paper are hand-crafted, the fact that experiments were conducted across a variety of styles suggests that this work holds sufficient value as a baseline.
> >
> > I would like to keep my score as it is.

---

> > > ### Author Response · Authors · 2026-04-07
> > >
> > > We thank **R-cGKm** for their insightful feedback and constructive discussion of our work. We are grateful for their positive assessment and pleased that our responses helped address the questions they raised.

---

### Decision · Program_Chairs · 2026-04-30

**Decision:**

Accept (spotlight)

**Comment:**

This paper considers reinforcement learning conditioned on styles. It first identifies three key challenges---(1) defining style; (2) distribution shift; and (3) resolving task/style misalignment---and then introduces Style-Conditioned Implicit Q-Learning (SCIQL) to address these challenges. SCIQL leverages sub-trajectory style labels and a gated advantage weighted regression mechanism to align optimized behavior with a particular style. The authors motivate this approach to address three key challenges: (1) defining style; (2) distribution shift; and (3) resolving task/style misalignment. Strong empirical results demonstrate the utility of the approach.

Though the reviewers raise some concerns over the burden of defining/annotating subtrajectory styles and choice of which challenge is emphasized in the paper, overall there is agreement that the paper is interesting and makes important contributions to this space.

I therefore recommend acceptance.